# Functional cross-talk between allosteric effects of activating and inhibiting ligands underlies PKM2 regulation

Jamie A Macpherson[1,2], Alina Theisen[3], Laura Masino[4], Louise Fets[1], Paul C Driscoll[5], Vesela Encheva[6], Ambrosius P Snijders[6], Stephen R Martin[4], Jens Kleinjung[7], Perdita E Barran[3], Franca Fraternali[2]*, Dimitrios Anastasiou[1]*

[1]Cancer Metabolism Laboratory, The Francis Crick Institute, London, United Kingdom; [2]Randall Centre for Cell and Molecular Biophysics, King's College London, London, United Kingdom; [3]Michael Barber Centre for Collaborative Mass Spectrometry, Manchester Institute of Biotechnology, School of Chemistry, University of Manchester, Manchester, United Kingdom; [4]Structural Biology Science Technology Platform, The Francis Crick Institute, London, United Kingdom; [5]Metabolomics Science Technology Platform, The Francis Crick Institute, London, United Kingdom; [6]Proteomics Science Technology Platform, The Francis Crick Institute, London, United Kingdom; [7]Computational Biology Science Technology Platform, The Francis Crick Institute, London, United Kingdom

*For correspondence:
franca.fraternali@kcl.ac.uk (FF);
dimitrios.anastasiou@crick.ac.uk (DA)

**Abstract** Several enzymes can simultaneously interact with multiple intracellular metabolites, however, how the allosteric effects of distinct ligands are integrated to coordinately control enzymatic activity remains poorly understood. We addressed this question using, as a model system, the glycolytic enzyme pyruvate kinase M2 (PKM2). We show that the PKM2 activator fructose 1,6-bisphosphate (FBP) alone promotes tetramerisation and increases PKM2 activity, but addition of the inhibitor L-phenylalanine (Phe) prevents maximal activation of FBP-bound PKM2 tetramers. We developed a method, AlloHubMat, that uses eigenvalue decomposition of mutual information derived from molecular dynamics trajectories to identify residues that mediate FBP-induced allostery. Experimental mutagenesis of these residues identified PKM2 variants in which activation by FBP remains intact but cannot be attenuated by Phe. Our findings reveal residues involved in FBP-induced allostery that enable the integration of allosteric input from Phe and provide a paradigm for the coordinate regulation of enzymatic activity by simultaneous allosteric inputs.

DOI: https://doi.org/10.7554/eLife.45068.001

## Introduction

Allostery refers to the regulation of protein function resulting from the binding of an effector to a site distal to the protein's functional centre (the active site in the case of enzymes) and is a crucial mechanism for the control of multiple physiological processes (*Shen et al., 2016*; *Nussinov and Tsai, 2013*). The functional response of enzymes to allosteric ligands occurs on the ns - ms time scale, preceding other important regulatory mechanisms such as changes in gene expression or signalling-induced post-translational modifications (PTMs) (*Gerosa and Sauer, 2011*), thereby enabling fast cellular responses to various stimuli. Despite the demonstrated importance of protein allostery, the investigation of underlying mechanisms remains a challenge for conventional structural approaches and necessitates multi-disciplinary strategies. Latent allosteric pockets can emerge as a

consequence of protein flexibility (*Bowman and Geissler, 2012*; *Keedy et al., 2018*) making their identification elusive. Even for known allosteric pockets, an understanding of the molecular mechanisms that underpin the propagation of free energy between an identified allosteric site and the active site can be complicated because of the involvement of protein structural motions on a variety of time scales (*Motlagh et al., 2014*). Furthermore, many proteins contain several allosteric pockets and can simultaneously bind more than one ligand (*Iturriaga-Vásquez et al., 2015*; *Macpherson and Anastasiou, 2017*; *Sieghart, 2015*). Concurrent binding of allosteric ligands can modulate the functional response of a protein through the action of multiple allosteric pathways (*del Sol et al., 2009*), however, it remains unclear whether such allosteric pathways operate independently, or integrate, either synergistically or antagonistically, to control protein function.

Altered allosteric regulation has a prominent role in the control of tumour metabolism, as well as a number of other pathological processes (*Nussinov and Tsai, 2013*; *Macpherson and Anastasiou, 2017*; *DeLaBarre et al., 2014*). Changes in glycolysis observed in some tumours have been linked to aberrant allosteric regulation of glycolytic enzymes including phosphofructokinase, triose phosphate isomerase and pyruvate kinase. Pyruvate kinases (PKs) catalyse the transfer of a phosphate from phosphoenolpyruvate (PEP) to adenosine diphosphate (ADP) and produce pyruvate and adenosine triphosphate (ATP). There are four mammalian isoforms of pyruvate kinase: PKM1, PKM2, PKL and PKR. PKM2 is highly expressed in tumour cells and in many proliferative tissues, and has critical roles in cancer metabolism that remain under intense investigation (*Allen and Locasale, 2018*; *Anastasiou et al., 2011*; *Christofk et al., 2008a*; *Dayton et al., 2016*), as well as in controlling systemic metabolic homeostasis and inflammation (*Palsson-McDermott et al., 2015*; *Xie et al., 2016*; *Qi et al., 2017*).

In contrast to the highly homologous alternatively spliced variant PKM1, which is thought to be constitutively active, PKM2 activity in cancer cells is maintained at a low level by the action of various allosteric ligands or post-translational modifications (PTMs) (*Chaneton and Gottlieb, 2012*). Low PKM2 activity promotes pro-tumorigenic functions, including divergence of glucose carbons into biosynthetic and redox-regulating pathways that support proliferation and defence against oxidative stress (*Anastasiou et al., 2011*; *Christofk et al., 2008a*; *Anastasiou et al., 2012*; *Lunt et al., 2015*). Small-molecule activators, which render PKM2 constitutively active by overcoming endogenous inhibitory cues, attenuate tumour growth, suggesting that regulation of PKM2 activity, rather than increased PKM2 expression per se, is important (*Anastasiou et al., 2011*; *Anastasiou et al., 2012*; *Wang et al., 2017a*; *Kim et al., 2015*). Therefore, PKM2 has emerged as a prototypical metabolic enzyme target for allosteric modulators and this has contributed to the renewed impetus to develop allosteric drugs for metabolic enzymes, which are expected to specifically interfere with cancer cell metabolism while sparing normal tissues (*DeLaBarre et al., 2014*).

The structure of the PKM2 protomer comprises N-terminal, A, B and C domains (*Figure 1*). Various ligands that regulate PKM2 activity bind to sites distal from the catalytic core, nestled between the A and B domains. The upstream glycolytic intermediate fructose 1,6-bisphosphate (FBP) binds to a pocket in the C-domain (*Dombrauckas et al., 2005*) and increases the enzymatic activity of PKM2, thereby establishing a feed-forward loop that prepares lower glycolysis for increased levels of incoming glucose carbons. Activation of PKM2 by FBP is associated with a decreased $K_M$ for the substrate PEP ($K_M^{PEP}$), while the $k_{cat}$ remains unchanged (*Chaneton et al., 2012*; *Boxer et al., 2010*; *Jiang et al., 2010*; *Yacovan et al., 2012*), although some reports also find an elevated $k_{cat}$ (*Morgan et al., 2013*; *Akhtar et al., 2009*) and the reason for this discrepancy remains unknown. Additionally, several amino acids regulate PKM2 activity by binding to a pocket in the A domain TIM-barrel core (*Chaneton et al., 2012*; *Yuan et al., 2018*; *Eigenbrodt et al., 1983*). L-serine (Ser) and L-histidine (His) increase, whereas L-phenylalanine (Phe), L-alanine (Ala), L-tryptophan (Trp), L-valine (Val) and L-proline (Pro) decrease $K_M^{PEP}$. Similar to FBP, the reported effects on $k_{cat}$ vary (*Chaneton et al., 2012*; *Morgan et al., 2013*; *Yuan et al., 2018*; *Gosalvez et al., 1975*). Furthermore, succinyl-5-aminoimidazole-4-carboxamide-1-ribose 5′-phosphate (SAICAR) (*Keller et al., 2012*) and the triiodothyronine ($T_3$) hormone (*Morgan et al., 2013*) bind to unidentified PKM2 pockets to increase and decrease, respectively, the affinity for PEP.

Many of these allosteric effectors have been shown to regulate PKM2 activity by changing the equilibrium of PKM2 between one of three states – a low activity monomer, a low activity tetramer (T-state), and a high activity tetramer (R-state) – with some studies reporting the existence of low activity dimers (*Dombrauckas et al., 2005*; *Morgan et al., 2013*; *Gavriilidou et al., 2018*;

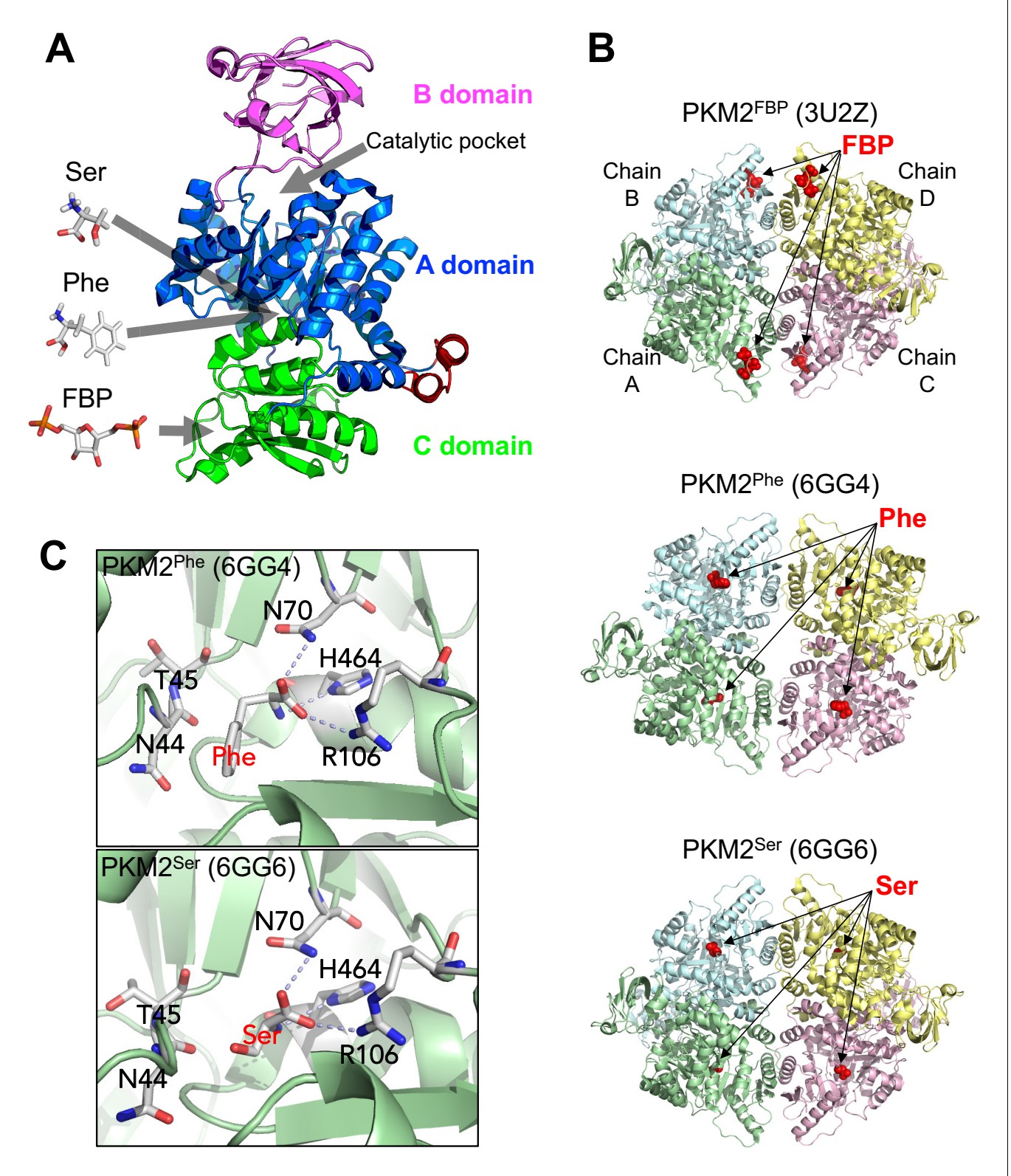

**Figure 1.** Overview of PKM2 structure and allosteric ligand binding sites. (**A**) Domain structure of PKM2 monomer (PDBID: 3BJT) depicting orthosteric and allosteric sites discussed in this work. (**B**) Published crystal structures of PKM2 bound to FBP, Phe or Ser (PDBID shown in parentheses). (**C**) Phe and Ser bind to the same pocket in PKM2. Close-up view of the Phe (top) and Ser (bottom) binding pockets with key contact residues [Asn(N)44, Thr(T)45, Asn(N)70, Arg(R)106, His(H)464] shown in stick configuration. Dashed lines indicate hydrogen bonds between PKM2 residues and Phe or Ser.

*Figure 1 continued on next page*

*Figure 1 continued*

DOI: https://doi.org/10.7554/eLife.45068.002

*Yan et al., 2016*; *Ashizawa et al., 1991a*; *Hofmann et al., 1975*; *Mazurek, 2011*). FBP promotes, whereas $T_3$ prevents, tetramerisation (*Dombrauckas et al., 2005*; *Morgan et al., 2013*; *Kato et al., 1989*). The mode of PKM2 regulation by amino acids is unclear. Some reports suggest that Ala promotes the formation of inactive PKM2 dimers (*Hofmann et al., 1975*; *Felíu and Sols, 1976*), whereas others show that Phe, Ala and Trp stabilise the T-state tetramer (*Morgan et al., 2013*; *Yuan et al., 2018*). In addition to allosteric effectors, PTMs can also influence the oligomerisation of PKM2 protomers, although the effects of PTMs on the enzyme mechanism remain elusive. Nevertheless, cellular PKM2 activity is often inferred from the oligomeric state PKM2 is found to adopt (*Anastasiou et al., 2012*; *Wang et al., 2017a*; *Wang et al., 2017b*; *Lim et al., 2016*; *Hitosugi et al., 2009*; *Lv et al., 2013*; *Christofk et al., 2008b*). Collectively, current evidence suggests that a link between enzyme activity and oligomerisation state exists and, while not well understood, it may play a role in PKM2 regulation.

In vitro, PKM2 can bind concurrently to multiple allosteric effectors that might either reciprocally influence each other's action or exert independent effects on enzymatic activity. A PKM2 mutant that cannot bind FBP can still be activated by Ser and, conversely, a mutant that abolishes Ser binding can be activated by FBP (*Chaneton et al., 2012*), suggesting that amino acids could work independently from FBP to regulate PKM2. However, inhibitory amino acids that bind to the same pocket as Ser fail to inhibit the enzyme in the presence of FBP indicating a dominant influence of the latter (*Yuan et al., 2018*; *Sparmann et al., 1973*). Similarly, FBP can overcome PKM2 inhibition by $T_3$ (*Kato et al., 1989*). FBP has also been shown to attenuate, but not completely prevent, inhibition of PKM2 by Ala (*Ashizawa et al., 1991b*). Together, these observations highlight PKM2 as a suitable model system to study how inputs from multiple allosteric cues are integrated to regulate enzymatic activity. Such insights may also have implications for metabolic regulation in vivo. Nevertheless, it remains unclear whether simultaneous binding of PKM2 to more than one ligands is functionally relevant in cells, because little is known about the intracellular concentrations of allosteric effectors relative to their binding affinities for PKM2.

Here, we first measure intracellular concentrations of key allosteric regulators and their respective affinities for PKM2, and provide evidence that, in cells, PKM2 is saturated with FBP. We then show that FBP binding in vitro alters the outcome of Phe binding on PKM2 oligomerisation and activity compared to apo-PKM2. Using a novel computational framework to predict residues implicated in allosteric signal transmission from molecular dynamics (MD) simulations, we identify a network of PKM2 residues that mediates allosteric activation by FBP. Intriguingly, mutagenesis of some of these residues interferes with the ability of Phe to hinder PKM2 regulation by FBP but does not perturb FBP–induced activation itself. These residues, therefore, integrate inputs from Phe in FBP-bound PKM2 and underlie the functional synergism between ligands that bind to distinct pockets in regulating PKM2 activity.

## Results

### Simultaneous binding of multiple ligands to PKM2 is relevant for its regulation in cells

To explore whether simultaneous binding of FBP and amino acids is likely to occur and therefore to be relevant for the regulation of PKM2 in cells, we assessed the fractional saturation of PKM2 bound to FBP, Phe and Ser, in proliferating cancer cells (*Figure 2*). To this end, we first measured the affinity of PKM2 for FBP, Phe and Ser in vitro, using fluorescence emission spectroscopy and microscale thermophoresis. Similar to previous reports (*Christofk et al., 2008a*; *Gavriilidou et al., 2018*) we found that purified recombinant PKM2 remained bound to FBP, to varying degrees (*Figure 2—figure supplement 1*), despite extensive dialysis, suggesting a high binding affinity. We, henceforth, used PKM2 preparations with <25% molar stoichiometry of residual FBP to PKM2 and refer to these preparations with no added ligands as PKM2[apo]*.

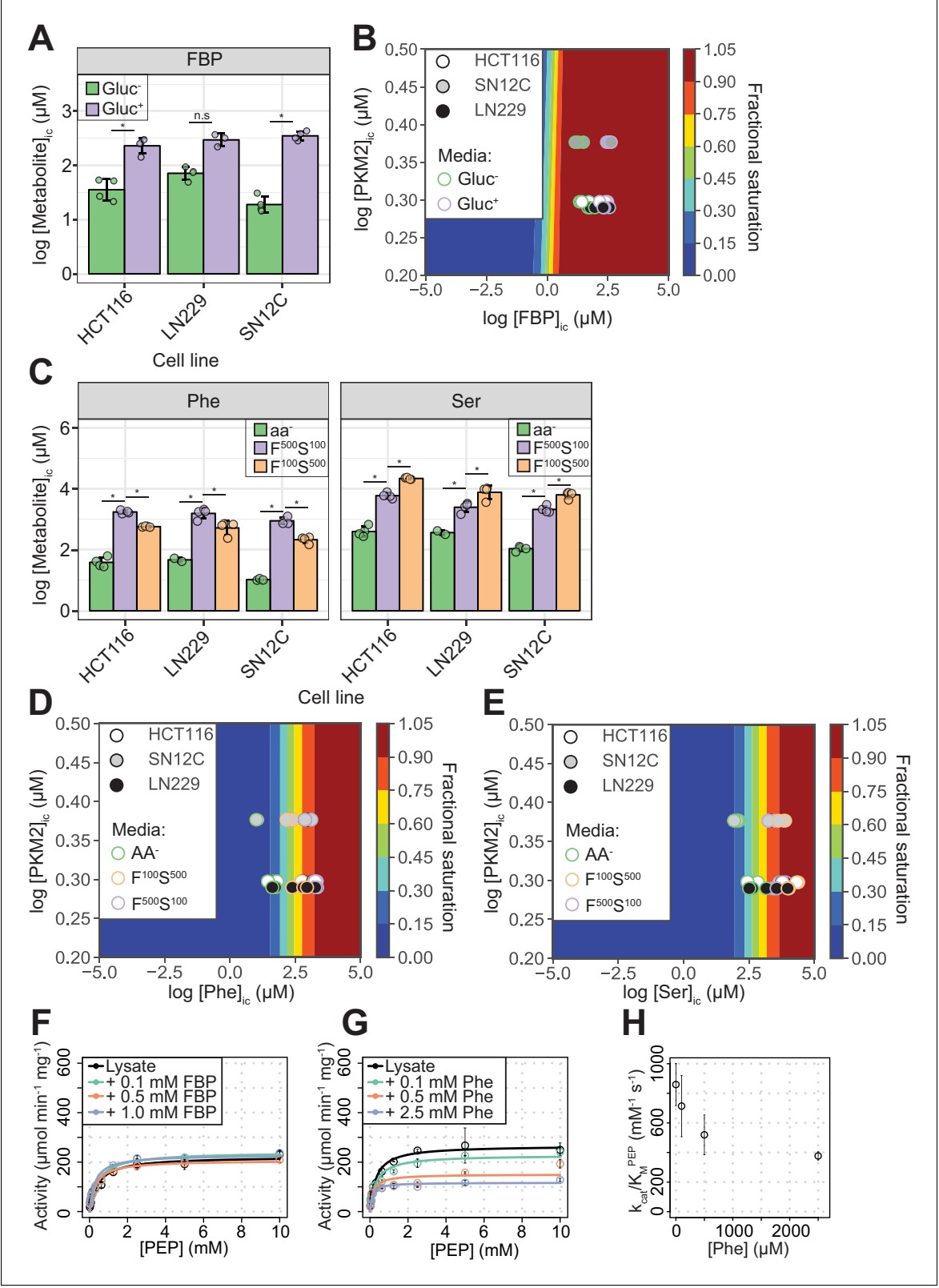

**Figure 2.** PKM2 allosteric effector concentrations in cells predict saturating binding of FBP and sub-saturating binding of Phe and Ser. (**A**) Intracellular concentrations of FBP ([FBP]$_{ic}$) measured using liquid-chromatography mass spectrometry (LC-MS), from HCT116 (colorectal carcinoma), LN229 (glioblastoma) and SN12C (renal cell carcinoma) cells cultured in RPMI media containing 11 mM glucose (Gluc$^+$), or 0 mM (Gluc$^-$) for 1 hr. Statistical significance was assessed using a Wilcoxon rank-sum test. Asterisk (*) marks significant changes (p-value<0.05). (**B**) Phase diagram for intracellular FBP

*Figure 2 continued on next page*

*Figure 2 continued*

binding to PKM2 computed for a range of [FBP] and [PKM2] values using 174 nM as the upper-limit estimate of the $K_D^{FBP}$, obtained as shown in *Figure 2—figure supplement 2*. Colour scale represents fractional saturation of PKM2 with ligand. A fractional saturation of 0 indicates no FBP bound to PKM2 and a fractional saturation equal to one indicates that each FBP binding site in the cellular pool of PKM2 would be occupied. Experimental fractional saturation values were estimated from $[FBP]_{ic}$ obtained from (A) and $[PKM2]_{ic}$ was determined using targeted proteomics (see Materials and methods and *Supplementary file 1*). The predicted fractional saturation for each of the three cell lines (four technical replicates) is shown as shaded open circles in the phase diagram. (C) Intracellular concentrations of Phe ($[Phe]_{ic}$) and Ser ($[Ser]_{ic}$) measured as in (A), in HCT116, LN229 and SN12C cells cultured in Hank's Balanced Salt Solution (HBSS) without amino acids ($aa^-$), HBSS containing 100 μM Phe and 500 μM Ser ($F^{100} S^{500}$), or HBSS containing 500 μM Phe and 100 μM Ser ($F^{500} S^{100}$). The low concentrations of Phe and Ser are similar to human serum concentrations (*Tardito et al., 2015*). $[Phe]_{ic}$ and $[Ser]_{ic}$ were not affected by extracellular glucose concentration (*Figure 2—figure supplement 4A*), neither did extracellular Phe and Ser concentrations influence $[FBP]_{ic}$ (*Figure 2—figure supplement 4B*). Statistical significance was assessed as in (A). (D) Phase diagram for Phe computed as in (B) using $[Phe]_{ic}$ from (C). (E) Phase diagram for Ser computed as in (B) using $[Ser]_{ic}$ from (C). (F) PKM2 activity in lysates of HCT116 cells cultured in RPMI ($Gluc^+$). Measurements were repeated following the addition of either 0.1, 0.5 or 1.0 mM of exogenous FBP. Initial velocity curves were fitted using Michaelis-Menten kinetics and the absolute concentration of PKM2 in the lysates was estimated using quantitative Western blotting (*Figure 2—figure supplement 5B*), to calculate PKM2 specific activity. (G) PKM2 activity in HCT116 cell lysates as in (F), but with addition of exogenous Phe. (H) Plot of $k_{cat}/K_M$ versus [Phe] from (G) revealing a dose-dependent inhibitory effect of Phe on the activity of PKM2 in HCT116 lysates.

DOI: https://doi.org/10.7554/eLife.45068.003

The following figure supplements are available for figure 2:

**Figure supplement 1.** Detection of residual FBP in purified recombinant PKM2 preparations.

DOI: https://doi.org/10.7554/eLife.45068.004

**Figure supplement 2.** FBP binds to PKM2 with nM affinity.

DOI: https://doi.org/10.7554/eLife.45068.005

**Figure supplement 3.** Affinities of Phe and Ser for PKM2.

DOI: https://doi.org/10.7554/eLife.45068.006

**Figure supplement 4.** Acute (1 hr) modulation of glucose and Phe/Ser concentration in the media does not affect intracellular concentrations of Phe/Ser or FBP, respectively.

DOI: https://doi.org/10.7554/eLife.45068.007

**Figure supplement 5.** PKM2 activity in cell lysates can be significantly modulated by exogenous amino acids but not by FBP.

DOI: https://doi.org/10.7554/eLife.45068.008

The $K_D^{FBP}$ was (25.5 ± 148.1) nM, after accounting (see Materials and methods) for the amount of co-purified FBP (*Figure 2—figure supplement 2A,B*). Owing to the high PKM2 concentration (5 μM) used in the assay relative to the measured affinity (see Materials and methods for details), the error in the estimated $K_D^{FBP}$ was substantial (*Figure 2—figure supplement 2B*). Nevertheless, the upper-limit estimate for the $K_D^{FBP}$ from ten replicate binding experiments was 174 nM, supporting the idea that FBP is a high-affinity ligand. The $K_D^{Phe}$ and $K_D^{Ser}$ were (191.0 ± 86.3) μM and (507.5 ± 218.2) μM, respectively (*Figure 2—figure supplement 3* and *Table 1*).

We next calculated the predicted fractional saturation range of PKM2 with FBP, Ser and Phe in cells. To this end, we determined the intracellular concentration of PKM2, using targeted proteomics, and the range of intracellular concentrations of FBP, Ser and Phe (denoted as $[X]_{ic}$, where X is the respective metabolite), using metabolomics, in three human cancer cell lines (see

**Table 1.** Apparent steady-state binding constants of PKM2 for FBP, Phe and Ser.

| Titrated ligand | Constant ligand | $K_D^{apparent}$ |
|---|---|---|
| FBP | – | (21.4 ± 9.0) nM |
| | Ser (5 mM) | (12.5 ± 7.7) nM |
| | Phe (1.5 mM) | (8.1 ± 5.6) nM |
| Phe | – | (191.0 ± 86.3) μM |
| | FBP (50 μM) | (132.0 ± 11.7) μM |
| Ser | – | (507.5 ± 218.2) μM |
| | FBP (50 μM) | (462.0 ± 97.1) μM |

DOI: https://doi.org/10.7554/eLife.45068.009

Materials and methods). In the presence of high (11 mM) extracellular glucose, $[FBP]_{ic}$ varied between 240–360 µM across the three cell lines ($Gluc^+$, **Figure 2A** and **Table 2**), a concentration range between 1380- and 2070-fold in excess of its binding affinity upper-limit estimate to PKM2. The calculated fractional saturation of PKM2 with FBP was 0.99 (**Figure 2B**) and remained unchanged even in cells cultured in the absence of glucose ($Gluc^-$), when $[FBP]_{ic}$ decreased to as low as 20 µM. In the presence of near-physiological extracellular concentrations of Phe or Ser ($Phe^{100}$ or $Ser^{100}$), $[Phe]_{ic}$ and $[Ser]_{ic}$ across the three lines ranged between 220–580 µM for Phe and 2000 - 6000 µM for Ser (**Figure 2C** and **Table 2**), close to their respective binding affinities for PKM2. The predicted fractional saturation range was 0.53–0.75 and 0.81–0.92 for Phe and Ser, respectively (**Figure 2D,E**). The range of predicted fractional saturations in complete absence of amino acids (-aa) or 5x physiological concentrations was 0.05 (-aa) to 0.90 ($Phe^{500}$) for Phe (**Figure 2D**) and 0.18 (-aa) to 0.98 ($Ser^{500}$) for Ser (**Figure 2E**). Under our experimental conditions, modulation of glucose concentration in media did not affect the concentrations of Phe or Ser, and *vice versa* (**Figure 2—figure supplement 4**). Together, these calculations predict that, in a range of physiologically relevant extracellular nutrient concentrations, FBP binding to PKM2 is near-saturating, whereas that of amino acids is not.

Consistent with the prediction of near-saturating PKM2 occupancy by FBP, addition of exogenous FBP to lysates of HCT116 cells cultured in both $Gluc^+$ and $Gluc^-$ media caused little increase in PKM2 activity (**Figure 2F** and **Figure 2—figure supplement 5A,B**). Addition of exogenous Phe (at physiological intracellular concentrations) to HCT116 lysates resulted in inhibition of PKM2 activity (**Figure 2G**) and a dose-dependent decrease in $k_{cat}/K_M$ (**Figure 2H**). Addition of exogenous Ser to HCT116 lysates reversed the inhibition of PKM2 activity by Phe, consistent with competitive binding between Ser and Phe for the same PKM2 pocket (**Yuan et al., 2018**) (**Figure 2—figure supplement 5B,C**).

Together, these observations support the concept that, during steady-state cell proliferation under typical culture conditions, a significant fraction of PKM2 is bound to FBP; they also suggest that amino acids can reversibly bind to and regulate PKM2 activity predominantly in the background of pre-bound FBP.

## Phe inhibits FBP–bound PKM2 without causing PKM2 tetramer dissociation

To investigate how allosteric ligands regulate PKM2, we measured their effects, first alone and then in combination, on PKM2 enzyme activity and oligomerisation. FBP activated PKM2 in a dose-dependent manner, with an apparent $AC_{50} = (118.1 \pm 19.0)$ nM (**Figure 3A**), consistent with the nM binding affinity estimate above. FBP decreased the $K_M^{PEP}$ to $(0.23 \pm 0.04)$ mM, compared to the absence of any added ligands ($PKM2^{apo*}$) [$K_M^{PEP} = (1.22 \pm 0.02)$ mM] (**Figure 3B** and **Table 3**). Similarly, addition of Ser resulted in a decrease of the $K_M^{PEP}$ to $(0.22 \pm 0.04)$ mM, whereas Phe increased the $K_M^{PEP}$ to $(7.08 \pm 1.58)$ mM. None of the three ligands changed the $k_{cat}$. These results are consistent with previous reports (**Dombrauckas et al., 2005**; **Chaneton et al., 2012**; **Ikeda and Noguchi, 1998**) that FBP, Phe and Ser change the $K_M^{PEP}$ but not the $k_{cat}$ and therefore act as K-type modulators (**Reinhart, 2004**) of PKM2.

In the presence of FBP, addition of Phe resulted in a decrease in the $k_{cat}$ of PKM2 from 349.3 $s^{-1}$ to 222.3 $s^{-1}$ (p=0.0075), and a simultaneous increase in the $K_M^{PEP}$ [$(0.65 \pm 0.03)$ mM] compared to $PKM2^{FBP}$ (**Figure 3B** and **Figure 3—figure supplement 1**). Further analysis of the kinetic data showed that in the absence of FBP, Phe acted as a hyperbolic-specific (**Baici, 2015**) inhibitor (**Figure 3—figure supplement 2A**), whereas, with FBP, Phe inhibition of PKM2 changed to a hyperbolic-mixed (**Baici, 2015**) mechanism (**Figure 3—figure supplement 2B**). In contrast, Ser caused no changes to either the $K_M^{PEP}$ or $k_{cat}$ in the presence of FBP (**Figure 3B**). Given that Phe and Ser bind to the same pocket (**Yuan et al., 2018**), we focused on Phe as an amino acid modulator of PKM2 for further investigations into the functional interaction between the amino acid and FBP binding pockets.

To explore the possibility that Phe modifies FBP binding and *vice versa*, we measured the binding affinities of each ligand alone, or in the presence of the other. Phe caused a small but not statistically significant (p=0.150) increase in the binding affinity of FBP ($K_D^{FBP}$) (**Figure 3C**), and, conversely, saturating amounts of FBP did not change the measured $K_D^{Phe}$ (**Figure 3D**). These measurements

**Table 2.** Intracellular concentrations of FBP, Phe and Ser in cancer cell lines.

| Metabolite | Media condition | Cell line | [Metabolite]$_{ic}$ ($\mu$M) | Fractional saturation |
|---|---|---|---|---|
| FBP | Gluc$^-$ (RPMI) | HCT116 | 38.4 ± 16.6 | 0.99 ± 0.00 |
| | | LN229 | 73.4 ± 18.4 | 0.99 ± 0.00 |
| | | SN12C | 19.9 ± 7.5 | 0.98 ± 0.00 |
| | Gluc$^+$ (RPMI) | HCT116 | 238.4 ± 67.4 | 0.99 ± 0.00 |
| | | LN229 | 302.0 ± 75.3 | 0.99 ± 0.00 |
| | | SN12C | 355.1 ± 66.2 | 0.99 ± 0.00 |
| | Gluc$^+$ aa$^-$ (HBSS) | HCT116 | 337.2 ± 185.5 | 0.99 ± 0.00 |
| | | LN229 | 257.0 ± 33.9 | 0.99 ± 0.00 |
| | | SN12C | 236.8 ± 41.1 | 0.99 ± 0.00 |
| | Gluc$^+$ Phe$^{500}$ Ser$^{100}$ (HBSS) | HCT116 | 384.5 ± 41.8 | 0.99 ± 0.00 |
| | | LN229 | 277.3 ± 83.4 | 0.99 ± 0.00 |
| | | SN12C | 300.3 ± 89.1 | 0.99 ± 0.00 |
| | Gluc$^+$ Phe$^{100}$ Ser$^{500}$ (HBSS) | HCT116 | 400.9 ± 82.2 | 0.99 ± 0.00 |
| | | LN229 | 372.2 ± 106.0 | 0.99 ± 0.00 |
| | | SN12C | 286.4 ± 73.6 | 0.99 ± 0.00 |
| Phe | Gluc$^-$ (RPMI) | HCT116 | 256.3 ± 65.8 | 0.56 ± 0.06 |
| | | LN229 | 231.6 ± 22.6 | 0.55 ± 0.03 |
| | | SN12C | 100.4 ± 24.6 | 0.34 ± 0.06 |
| | Gluc$^+$ (RPMI) | HCT116 | 258.2 ± 79.6 | 0.56 ± 0.08 |
| | | LN229 | 318.7 ± 115.8 | 0.61 ± 0.09 |
| | | SN12C | 197.5 ± 26.6 | 0.51 ± 0.04 |
| | Gluc$^+$ aa$^-$ (HBSS) | HCT116 | 41.1 ± 15.8 | 0.17 ± 0.05 |
| | | LN229 | 31.7 ± 28.0 | 0.13 ± 0.12 |
| | | SN12C | 10.6 ± 0.6 | 0.05 ± 0.02 |
| | Gluc$^+$ Phe$^{500}$ Ser$^{100}$ (HBSS) | HCT116 | 1776.2 ± 225.1 | 0.90 ± 0.01 |
| | | LN229 | 1666.5 ± 543.0 | 0.89 ± 0.04 |
| | | SN12C | 926.1 ± 252.7 | 0.82 ± 0.04 |
| | Gluc$^+$ Phe$^{100}$ Ser$^{500}$ (HBSS) | HCT116 | 582.4 ± 23.8 | 0.75 ± 0.01 |
| | | LN229 | 575.7 ± 224.4 | 0.73 ± 0.11 |
| | | SN12C | 221.6 ± 49.1 | 0.53 ± 0.06 |
| Ser | Gluc$^-$ (RPMI) | HCT116 | 3580.9 ± 1016.9 | 0.87 ± 0.03 |
| | | LN229 | 1755.0 ± 159.5 | 0.77 ± 0.02 |
| | | SN12C | 854.3 ± 240.7 | 0.62 ± 0.07 |
| | Gluc$^+$ (RPMI) | HCT116 | 3943.5 ± 1363.1 | 0.88 ± 0.05 |
| | | LN229 | 2226.7 ± 757.7 | 0.80 ± 0.06 |
| | | SN12C | 1766.0 ± 142.9 | 0.78 ± 0.01 |
| | Gluc$^+$ aa$^-$ (HBSS) | HCT116 | 426.9 ± 179.2 | 0.44 ± 0.09 |
| | | LN229 | 249.6 ± 221.8 | 0.28 ± 0.25 |
| | | SN12C | 111.0 ± 18.4 | 0.18 ± 0.02 |
| | Gluc$^+$ Phe$^{500}$ Ser$^{100}$ (HBSS) | HCT116 | 6157.3 ± 1334.4 | 0.92 ± 0.01 |
| | | LN229 | 2641.1 ± 825.1 | 0.83 ± 0.06 |
| | | SN12C | 2197.7 ± 605.8 | 0.81 ± 0.04 |
| | Gluc$^+$ Phe$^{100}$ Ser$^{500}$ (HBSS) | HCT116 | 22360.4 ± 1554.2 | 0.98 ± 0.00 |
| | | LN229 | 8464.7 ± 3214.5 | 0.93 ± 0.04 |
| | | SN12C | 6603.9 ± 1500.4 | 0.93 ± 0.02 |

DOI: https://doi.org/10.7554/eLife.45068.010

suggested that decreased binding of FBP could not account for the attenuation of FBP–induced activation of PKM2 upon addition of Phe.

We next investigated whether Phe impedes FBP–induced activation of PKM2 by perturbing the oligomeric state of the protein, using nano-electrospray ionisation (nESI) and ion mobility (IM) mass spectrometry (MS) (*Pacholarz et al., 2017*; *Beveridge et al., 2016*). The mass spectrum of PKM2$^{apo*}$ showed a mixture of monomers, dimers and tetramers at an approximate ratio of 1:7:10 (*Figure 4A, B*), with dimers exclusively forming the A-A' dimer assembly, as evidenced from experimental (IM) and theoretical measurements (*Figure 4—figure supplement 1*). Mass-deconvolution of the spectrum of PKM2$^{apo*}$ indicated that the signal of the tetramer consisted of five distinct species, which were assigned to PKM2$^{apo}$, and PKM2 bound to 1, 2, 3 or 4 molecules of FBP (*Figure 4C*), consistent with our earlier finding that FBP co-purifies with PKM2 (*Figure 2—figure supplement 1*). Addition of exogenous FBP resulted in a decrease in the intensity of monomer and dimer peaks, and a dose-dependent increase in the intensity of the tetramer signal (*Figure 4—figure supplement 2*). At saturating concentrations of FBP, the mass spectrum of PKM2 consisted of a single species corresponding to tetrameric PKM2 bound to four molecules of FBP (*Figure 4C*). Furthermore, surface-induced dissociation (SID) experiments revealed that PKM2$^{FBP}$ tetramers were more stable than PKM2$^{apo*}$ tetramers with respect to dissociation into dimers and monomers (*Figure 4—figure supplement 3*). Together, these data indicated that addition of FBP promotes the formation of PKM2 tetramers, as seen previously using other methods (*Ashizawa et al., 1991a*; *Kato et al., 1989*), conferring an enhanced tetramer stability.

Addition of Phe to PKM2$^{apo*}$ caused a decrease in the relative abundance of the tetramers and an increased relative abundance of the dimer species (*Figure 4A,B*). In contrast, Phe did not perturb the tetrameric state of PKM2$^{FBP}$ (*Figure 4A,B*), while FBP binding to PKM2$^{Phe}$ resulted in tetramerisation of the protein, indicating that the dominant effect of FBP on PKM2 tetramerisation was unaffected by the order of ligand addition (*Figure 4A,B*). PKM2 tetramers were found to bind FBP and Phe simultaneously, on the basis of the mass shift resulting from sequential addition of either ligand alone or in combination (*Figure 4—figure supplement 4*). Additionally, differences in the collision cross section ($^{DT}$CCS$_{He}$) distribution, which reflects the conformational heterogeneity of proteins, suggested that FBP caused subtle conformational changes in the tetramer, evidenced by a modest decrease in the $^{DT}$CCS$_{He}$ between 9500 Å$^2$–9650 Å$^2$ (*Figure 4D*). The addition of Phe partially reversed the changed $^{DT}$CCS$_{He}$ to a distribution closely resembling that of PKM2$^{apo*}$ (*Figure 4D*). Moreover, in the presence of Phe, half-stoichiometric amounts of FBP were sufficient to induce tetramerisation with slow kinetics [$k_{tet}$ = (812.5 s $\pm$ 284.6 s$^{-1}$)], whereas equivalent half-stoichiometric amounts of FBP in the absence of Phe were unable to fully convert PKM2 monomers and dimers into tetramers (*Figure 4E*). The propensity for Phe to enhance FBP-induced tetramerisation indicates a functional synergism between the two allosteric ligands, that favours tetramer formation despite the opposing effects of these ligands, individually, both on activity and oligomerisation.

Together, these data demonstrate that the inhibitory effect of Phe on the ability of FBP to enhance PKM2 activity is not due to Phe preventing FBP-induced tetramerisation and suggest it is likely due to interference with the allosteric communication between the FBP and the active site.

## Molecular dynamics simulations reveal candidate residues that mediate FBP–induced PKM2 allostery

To gain insight into the mechanism by which Phe interferes with FBP-induced allosteric activation of PKM2, we first sought to identify PKM2 residues that mediate the allosteric communication between the FBP binding site and the catalytic center. To this end, we performed molecular dynamics (MD) simulations of PKM2$^{apo}$ tetramers and PKM2$^{FBP}$ tetramers (*Table 6*). Subsequent analyses of protein volume and solvent accessibility of the trajectories (*Figure 5—figure supplement 1*), found no evidence of large global protein conformational changes induced by FBP, consistent with the modest $^{DT}$CCS$_{He}$ changes we observed by IM-MS (*Figure 4D*). We therefore reasoned that enthalpic motions likely play a role in the allosteric regulation of PKM2 by FBP.

To test this hypothesis, we set out to identify whether FBP elicits correlated motions (*Pandini et al., 2012*) in the backbone of PKM2, and whether these concerted motions form

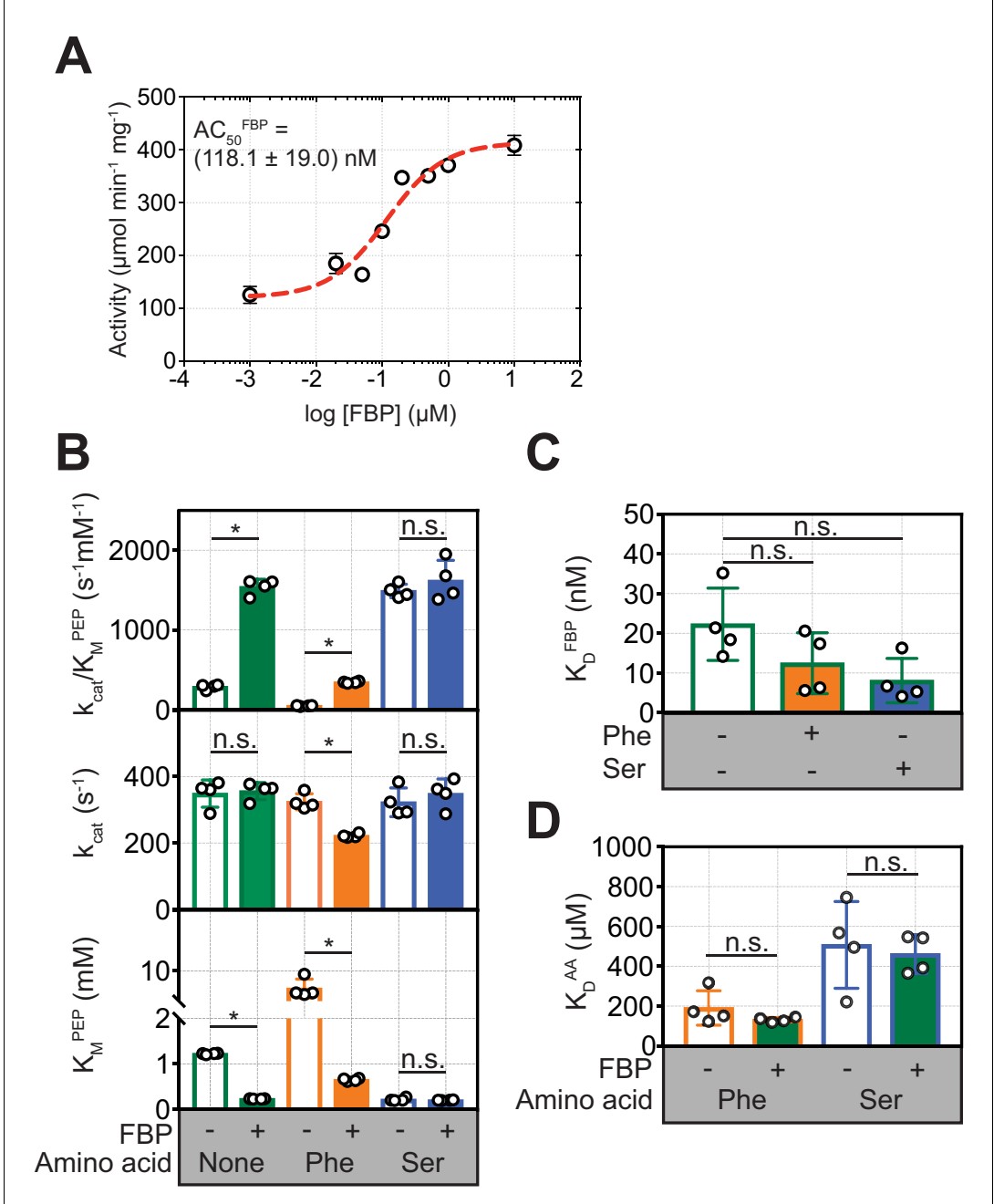

**Figure 3.** FBP influences the inhibition of PKM2 activity by Phe. (A) PKM2 (5 nM) activity measured over a range of added FBP concentrations (0.01–10 μM) with constant substrate concentrations (PEP = 1.5 mM and ADP = 5 mM). The apparent activation constant ($AC_{50}^{FBP}$) was estimated by fitting the resulting data to a binding curve (red line) assuming a 1:1 stoichiometry. Means and standard deviations of four separate experiments are plotted. (B) Steady-state kinetic parameters of purified recombinant human PKM2, in the absence of added ligands; in the presence of 2 μM FBP, 400 μM Phe and 200 mM Ser alone; and after addition of either 400 μM Phe or 200 mM Ser to PKM2 pre-incubated with 2 μM FBP. Initial velocity curves were fit to Michaelis-Menten kinetic models. Each titration was repeated four times. Statistical significance was assessed using a Wilcoxon rank-sum test. Asterisk (*) marks significant changes (p-value<0.05). (C) Binding constant of FBP to PKM2 obtained from fluorescence emission spectroscopy measurements in the absence and in the presence of either 400 μM Phe or 200 mM Ser. (D) Binding constants for Phe and Ser to PKM2, in the absence or presence of 2 μM FBP, obtained from microscale thermophoresis (MST) measurements. MST was used, rather than intrinsic fluorescence spectroscopy as for FBP, due to the absence of tryptophan residues proximal to the amino acid binding pocket on PKM2. Significance was assessed as in (A).
DOI: https://doi.org/10.7554/eLife.45068.012

The following figure supplements are available for figure 3:

**Figure supplement 1.** FBP influences the kinetics of PKM2 inhibition by Phe - experimental data.

*Figure 3 continued on next page*

*Figure 3 continued*

DOI: https://doi.org/10.7554/eLife.45068.013

**Figure supplement 2.** FBP influences the kinetics of PKM2 inhibition by Phe - kinetic mechanism models.

DOI: https://doi.org/10.7554/eLife.45068.014

the basis of a network of residues that connect the allosteric pocket to the active site. Accurately computing the network of correlated protein motions from MD simulations is complicated by the occurrence of dynamic conformational sub-states that can display unique structural properties (*Guerry et al., 2013*; *Bürgi et al., 2001*). We therefore developed a novel computational framework, named *AlloHubMat* (**Allo**steric **Hub** prediction using **Mat**rices that capture allosteric coupling), to predict allosteric hub fragments from the network of dynamic correlated motions, based on explicitly identified conformational sub-states from multiple MD trajectories (*Figure 5A*). Extraction of correlated motions from multiple sub-states within a consistent information-theoretical framework allowed us to compare the allosteric networks, both between replicas of the same liganded state and between different liganded states of PKM2 (see Materials and methods).

Using AlloHubMat, we analysed all replicate MD simulations of PKM2$^{apo}$ and PKM2$^{FBP}$ and identified several conformational sub-states. Backbone correlations extracted from the sub-states separated into two clusters (C1 and C2) that were dominated by sub-states from PKM2$^{apo}$ and PKM2$^{FBP}$ simulations, respectively, in addition to cluster C3 that was populated by sub-states from both simulations (*Figure 5B*). The observed separation in the correlated motions revealed common conformational sub-states, suggesting that the preceding analysis of MD simulations of PKM2 captured FBP–dependent correlated motions.

To identify allosteric hub fragments (AlloHubFs) that are involved in the allosteric state transition, we subtracted the mutual information matrices identified in PKM2$^{apo}$ from those in PKM2$^{FBP}$ (*Figure 5C*). We found that the strength of the coupling signal between the AlloHubFs correlated with the positional entropy (*Figure 5D*), corroborating the idea that local backbone flexibility contributed to the transmission of allosteric information. The top ten predicted AlloHubFs (named Hub$_1$-Hub$_{10}$) were spatially dispersed across the PKM2 structure including positions proximal to the A-A' interface (Hub$_5$ and Hub$_6$), to the FBP binding pocket (Hub$_9$), the C-C' interface (Hub$_{10}$), and within the B-domain (Hub$_1$ and Hub$_2$) (*Figure 5E*). Remarkably, all AlloHubFs, with the exception of Hub$_5$ and Hub$_6$, coincided with minimal distance pathways (*Dijkstra, 1959*) between the FBP binding pocket and the active site (*Figure 5E*). This observation further supported the hypothesis that the selected AlloHubFs propagate the allosteric effect of FBP.

## AlloHubF mutants disrupt FBP-induced activation of PKM2 or its sensitivity to Phe

We next generated *allosteric hub mutants* (AlloHubMs) (*Figure 6—figure supplement 1*) by substituting AlloHubF residues with amino acids that had chemically different side chains and were predicted to be tolerated at the respective position based on their occurrence in a multiple sequence alignment of 5381 pyruvate kinase orthologues (*Figure 6—figure supplement 2*). Among a total of 23 PKM2 AlloHubMs generated, we chose seven [I124G, F244V, K305Q, F307P, A327S, C358A,

**Table 3.** Steady-state Michaelis-Menten kinetic parameters for PKM2$^{apo}$, and following the addition of FBP, Phe and Ser.

| PKM2$^{ligand}$ | $K_M^{PEP}$ (mM) | $k_{cat}$ (s$^{-1}$) | $k_{cat}/K_M^{PEP}$ (s$^{-1}$ mM$^{-1}$) |
|---|---|---|---|
| PKM2$^{apo}$ | 1.22 ± 0.02 | 349.3 ± 40.9 | 285.6 ± 34.1 |
| PKM2$^{Phe}$ | 7.08 ± 1.58 | 324.7 ± 23.9 | 46.8 ± 6.0 |
| PKM2$^{Ser}$ | 0.22 ± 0.04 | 323.1 ± 43.2 | 1489.8 ± 84.7 |
| PKM2$^{FBP}$ | 0.23 ± 0.04 | 356.7 ± 25.7 | 1540.4 ± 96.9 |
| PKM2$^{FBP + Phe}$ | 0.65 ± 0.03 | 222.3 ± 6.3 | 342.1 ± 11.1 |
| PKM2$^{FBP + Ser}$ | 0.20 ± 0.04 | 348.7 ± 44.3 | 1620.0 ± 253.6 |

DOI: https://doi.org/10.7554/eLife.45068.011

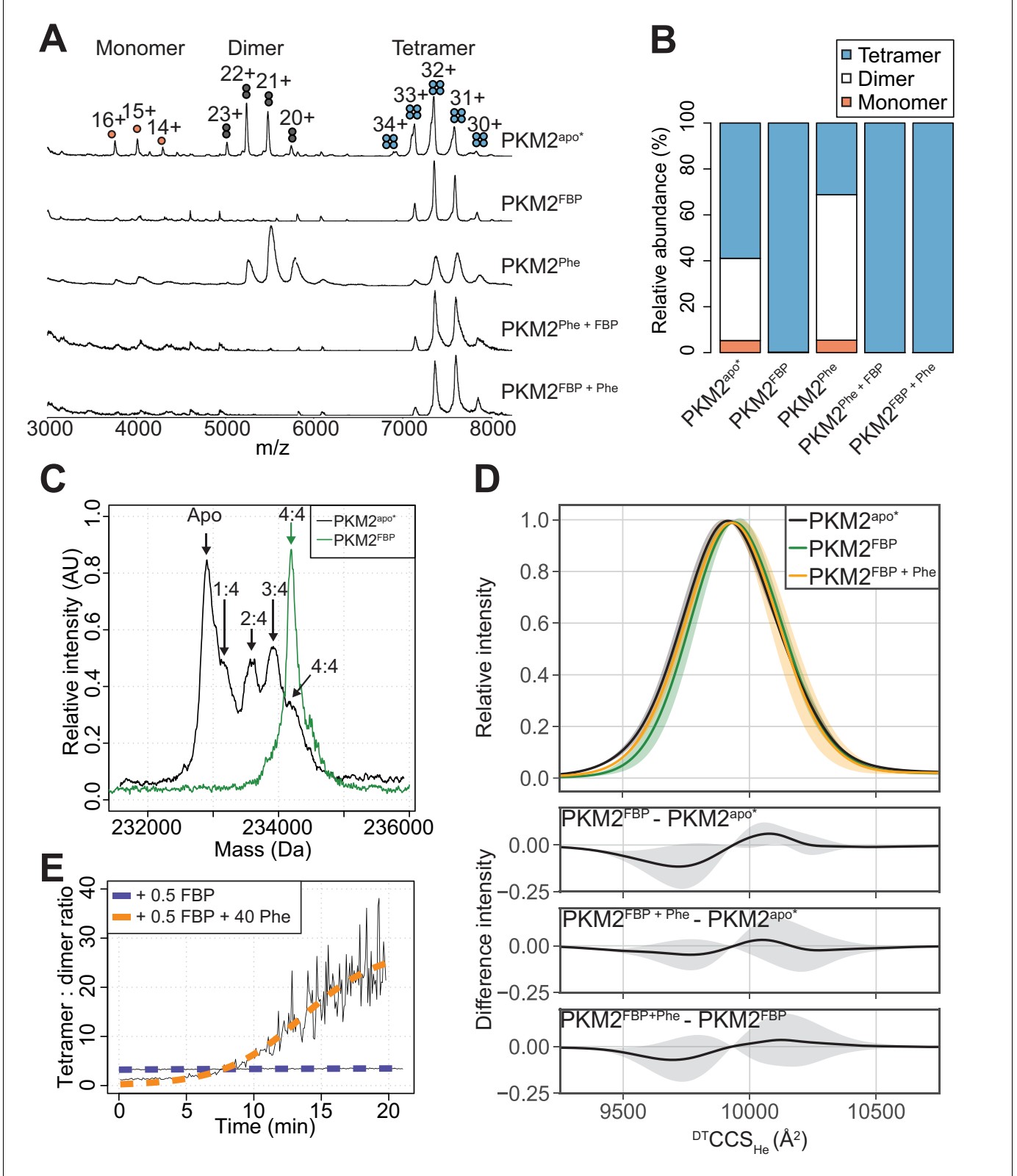

**Figure 4.** FBP modulates the effects of Phe on PKM2 oligomerisation. (A) Native mass spectra of 10 µM PKM2 in 200 mM ammonium acetate at pH 6.8, in the absence of allosteric ligands (PKM2[apo*]), or in the presence of: 10 µM FBP (PKM2[FBP]), 300 µM Phe (PKM2[Phe]), 300 µM Phe followed by addition of 10 µM FBP (PKM2[Phe + FBP]) or 10 µM FBP followed by addition of 300 µM Phe (PKM2[FBP + Phe]). (B) Relative abundance of PKM2 monomers, dimers and tetramers obtained from the spectra shown in (A) by computing the area of the peaks corresponding to each of the three oligomeric states. Relative

*Figure 4 continued on next page*

*Figure 4 continued*

peak areas were calculated as a percentage of the total area given by all charge-state species in a single mass spectrum. (C) Deconvolved mass spectra of PKM2 tetrameric species in the absence of any added ligands (PKM2$^{apo*}$) or presence of FBP (PKM2$^{FBP}$). PKM2$^{apo*}$ has five distinct mass peaks, separated by approximately 340 Da (equivalent to the weight of FBP), corresponding to tetrameric PKM2$^{apo*}$, and tetrameric PKM2 bound to 1, 2, 3 and 4 molecules of FBP, respectively. See *Table 4* for the theoretical, and *Table 5* for the experimentally measured masses of PKM2, FBP and their complexes. The spectrum of PKM2$^{FBP}$ contains a single peak, corresponding to tetrameric PKM2 bound to four molecules of FBP. (D) $^{DT}CCS_{He}$ distribution of PKM2$^{apo*}$, PKM2$^{FBP}$ and PKM2$^{FBP+Phe}$ tetramers calculated from analyses of arrival time distribution measurements of PKM2 tetramer peaks (see Materials and methods). The plots at the bottom show the mean difference of the $^{DT}CCS_{He}$ distributions between the indicated liganded states. Grey shaded regions show the standard deviations of the distribution differences. (E) Change, over time, in the oligomeric state of PKM2 upon addition of sub-stoichiometric FBP and saturating Phe concentrations. Oligomerisation is reported as the ratio of the tetramer 32 + charge state peak relative to the dimer 22 + charge state peak, obtained from mass spectra of 10 µM PKM2 following addition of either 5 µM FBP, or 5 µM FBP and 400 µM Phe over the course of 20 min. The kinetics of tetramerisation were estimated from a two-state sigmoidal model (orange and blue dashed lines, see Materials and methods). In the legend, 0.5 FBP and 40 Phe indicate molar ratio of these ligands compared to PKM2.

DOI: https://doi.org/10.7554/eLife.45068.015

The following figure supplements are available for figure 4:

**Figure supplement 1.** Evidence from IM-MS and MD simulations that PKM2 dimers predominantly adopt the A-A' configuration.
DOI: https://doi.org/10.7554/eLife.45068.016
**Figure supplement 2.** FBP promotes a dose-dependent conversion of PKM2 monomers and A-A' dimers into the tetrameric species.
DOI: https://doi.org/10.7554/eLife.45068.017
**Figure supplement 3.** FBP binding increases the stability of PKM2 tetramers.
DOI: https://doi.org/10.7554/eLife.45068.018
**Figure supplement 4.** Phe and FBP can simultaneously bind to PKM2.
DOI: https://doi.org/10.7554/eLife.45068.019

R489L (*Figure 6A*)] for further experimental characterization because they expressed as soluble proteins and had very similar secondary structure to that of PKM2(WT), which suggested that the protein fold in these mutants was largely preserved (*Figure 6—figure supplement 3*). Importantly, the $K_D^{FBP}$ for all AlloHubMs were similar to PKM2(WT), with the exception of PKM2(R489L), which bound to FBP with low affinity (*Figure 6—figure supplement 4* and *Table 7*).

In order to quantify and compare the ability of FBP to activate AlloHubMs independently of varying basal activity, we determined the allosteric coupling constant (*Reinhart, 1983*; *Carlson and Fenton, 2016*,) $\log_{10}Q^{FBP}$, defined as the log-ratio of the $K_M^{PEP}$ in the absence over the $K_M^{PEP}$ in the presence of FBP. PKM2 AlloHubMs I124G, F244V, K305Q, F307P and R489L showed attenuated activation by FBP (*Figure 6B*) indicating that AlloHubMat successfully identified residues that mediate the allosteric effect of FBP. Similar to PKM2(WT), Phe addition significantly hindered FBP-induced activation in I124G, F244V, and R489L (*Figure 6C*). In contrast, K305Q and F307P were allosterically inert, with no detectable response in activity upon addition of either FBP or Phe (*Figure 6B,C* and *Figure 6—figure supplement 5*). While for PKM2(K305Q) this outcome could be explained by very low basal activity, PKM2(F307P) had a $K_M^{PEP}$ similar to that of PKM2(WT)$^{FBP}$, indicating that this mutant displays a constitutively high substrate affinity (*Figure 6—figure supplement 5* and *Table 8*).

**Table 4.** Theoretical masses of PKM2 (see *Supplementary file 2* for PKM2 sequence), FBP and PKM2 + FBP.

| Molecule | Mass (Da) |
| --- | --- |
| PKM2 monomer | 58218.2 |
| PKM2 tetramer | 232872.6 |
| FBP | 340.0 |
| PKM2 tetramer + 1 FBP | 233212.6 |
| PKM2 tetramer + 2 FBP | 233552.6 |
| PKM2 tetramer + 3 FBP | 233892.6 |
| PKM2 tetramer + 4 FBP | 234232.6 |

DOI: https://doi.org/10.7554/eLife.45068.020

**Table 5.** Calculated masses from maximum-entropy deconvolution of PKM2 mass spectra.

| Protein/ligand mixture | Cone voltage (V) | Number of tetramer peaks | Mass species (Da) |
|---|---|---|---|
| 10 µM PKM2 | 100 | 5 | 232880, 233220, 233580, 233930, 234290 |
| 10 µM PKM2 + 10 µM FBP | 10 | 1 | 235030 |
| 10 µM PKM2 + 10 µM FBP | 100 | 1 | 234190 |

DOI: https://doi.org/10.7554/eLife.45068.021

The mass spectra of the AlloHubMs (*Figure 6D*) revealed a marked decrease in the intensity of tetramer and dimer peaks for PKM2(K305Q) and PKM2(F307P) compared to PKM2(WT), consistent with the position of these residues on the A-A' interface, along which stable PKM2(WT) dimers are formed (*Figure 4—figure supplement 1*). However, upon addition of FBP, F307P remained largely monomeric, whereas K305Q formed tetramers with a similar charge state distribution to PKM2(WT)$^{apo*}$ (*Figure 6—figure supplement 6*). These observations, in addition to the varying ability of I124G, F244V, and R489L to tetramerise upon addition of FBP (I124G > F244V>R489L, *Figure 6—figure supplement 6*), indicate that an impaired allosteric activation of these mutants by FBP cannot be accounted for by altered oligomerisation alone.

Intriguingly, two AlloHubMs, PKM2(A327S) and PKM2(C358A), retained intact activation by FBP (*Figure 6B*), suggesting either that the amino acid substitutions were functionally neutral or that these residues are not required for FBP-induced activation. However, addition of Phe failed to attenuate FBP-induced activation of these AlloHubMs (*Figure 6C*), indicating that residues A327 and C358 have a role in coupling the allosteric effect of Phe with that of FBP.

In summary, evaluation of the allosteric properties of AlloHubMs demonstrated that AlloHubMat successfully identified residues involved in FBP-induced allosteric activation of PKM2. Furthermore, this analysis revealed two residues that mediate a functional cross-talk between allosteric networks elicited from distinct ligand binding pockets on PKM2, thereby providing a mechanism by which distinct ligands synergistically control PKM2 activity.

## Discussion

Allosteric activation of PKM2 by FBP is a prototypical and long-studied example of feed-forward regulation in glycolysis (*Koler and Vanbellinghen, 1968*). However, PKM2 binds to many other ligands in addition to FBP, including inhibitory amino acids. It has been unclear, thus far, whether ligands that bind to distinct pockets elicit functionally independent allosteric pathways to control PKM2 activity or whether they synergise, and if so, which residues mediate such synergism. Furthermore, the role played by simultaneous binding of multiple ligands on the oligomerisation state of PKM2 remained elusive. Our work shows that FBP-induced dynamic coupling between distal residues functions, in part, to enable Phe to interfere with FBP-induced allostery. This finding points to a functional cross-talk between the allosteric mechanisms of these two ligands.

### AlloHubMat reveals residues that mediate a cross-talk between FBP- and Phe-induced allosteric regulation

Multiple lines of evidence suggested a functional cross-talk between the allosteric mechanisms of Phe and FBP. While Phe and FBP bind to spatially distinct pockets on PKM2, both ligands influence the mode of action of the other, without reciprocal effects on their binding affinities. Using native

**Table 6.** Summary of molecular dynamics simulations.

| Liganded state simulated | PDB ID of starting structure | Number of protein atoms | Number of water atoms | Simulation time | Number of replicas |
|---|---|---|---|---|---|
| PKM2$^{apo}$ | 3BJT | 20104 | 263384 | 400 ns | 5 |
| PKM2$^{FBP}$ | 3U2Z | 20122 | 261594 | 420 ns | 5 |

DOI: https://doi.org/10.7554/eLife.45068.022

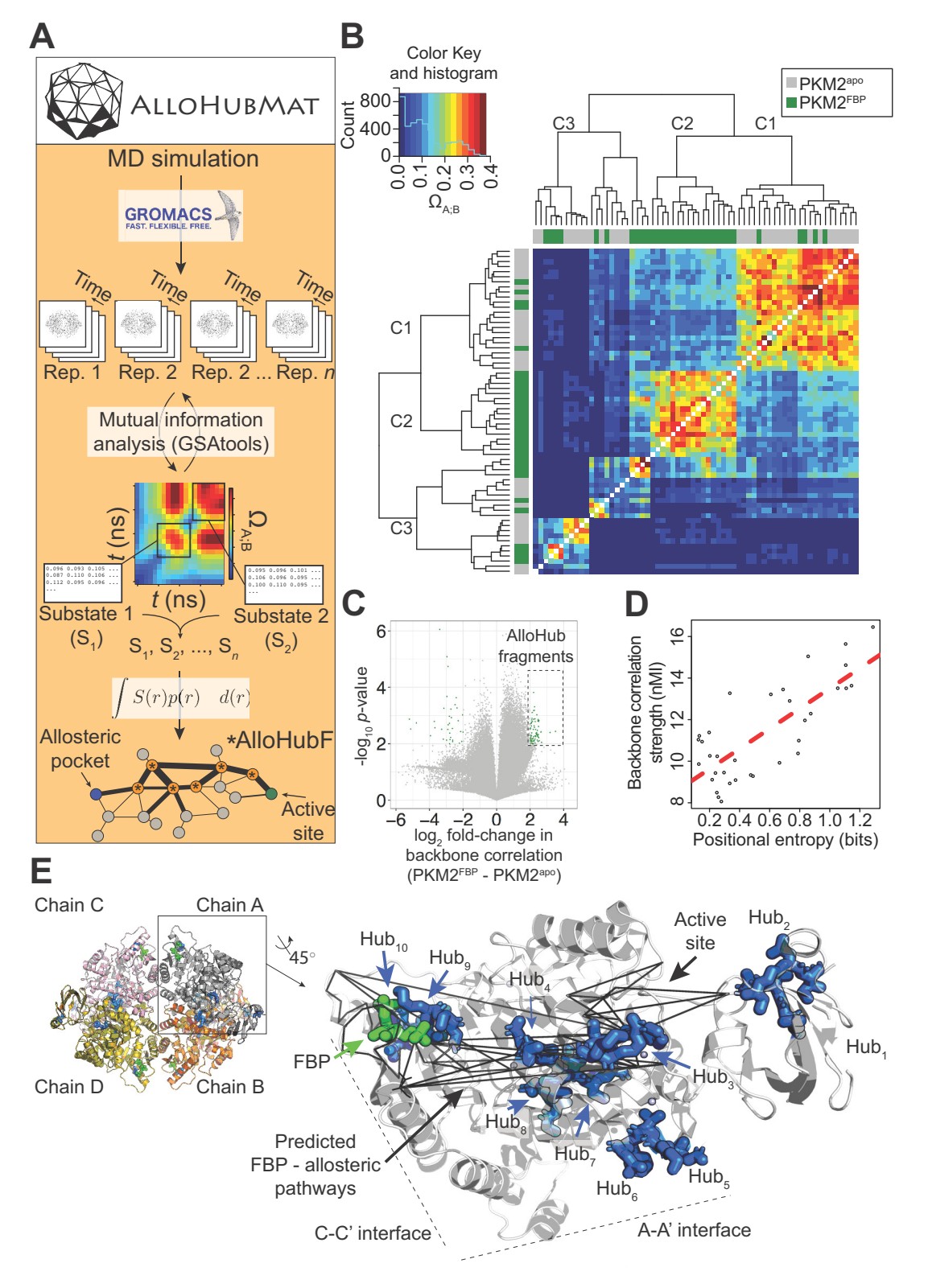

**Figure 5.** AlloHubMat predicts candidate residues that mediate the allosteric effect of FBP on PKM2, from molecular dynamics (MD) simulations. (**A**) Schematic of the AlloHubMat computational pipeline, developed to identify residues that are involved in the transmission of allostery between an allosteric ligand pocket and the active site. Multiple replicate molecular dynamics (MD) simulations are seeded from the 3D protein structure using the GROMACS molecular dynamics engine. All MD simulations are encoded with the M32K25 structural alphabet (*Pandini et al., 2010*), and the protein

*Figure 5 continued on next page*

*Figure 5 continued*

backbone correlations over the MD trajectory are computed with GSAtools (*Pandini et al., 2013*) using information theory mutual information statistics. The backbone correlations are explicitly used to identify and extract configurational sub-states from the MD trajectories. A global allosteric network is then constructed by integrating over the correlation matrices, and their respective probabilities, from which allosteric hub fragments (AlloHubFs) are extracted. Each AlloHubF comprises four consecutive amino acid residues. (B) Correlation matrices cluster according to the liganded state of PKM2 in the MD simulations. AlloHubMat, described in (A), was used to identify correlation matrices of the conformational substates from five separate 400 ns MD simulations of PKM2$^{apo}$ (grey) and PKM2$^{FBP}$ (green). In total, we identified eight sub-states for all simulations of PKM2$^{apo}$ and eight for PKM2$^{FBP}$. For every sub-state, the network of correlations from each of the four protomers is presented individually. To investigate whether the correlated motions for each sub-state could be attributed to the liganded state of PKM2, the correlation matrices were compared with a complete-linkage hierarchical clustering (see Materials and methods). The matrix covariance overlap ($\Omega_{A;B}$) was used as a distance metric, represented by the colour scale. A high $\Omega_{A;B}$ score indicates high similarity between two correlation matrices, and a low $\Omega_{A;B}$ score indicates that the correlation matrices are dissimilar. The clustering analysis revealed three clusters, denoted C1-C3. Cluster C1 was dominated by correlation matrices extracted from PKM2$^{apo}$ simulations, and cluster C2 was exclusively occupied by PKM2$^{FBP}$ correlation matrices. C3 consisted of correlation matrices from PKM2$^{apo}$ and PKM2$^{FBP}$ simulations. (C) A volcano plot showing difference in protein backbone correlations – derived from the AlloHubMat analysis – between PKM2$^{apo}$ and PKM2$^{FBP}$. Each point corresponds to a correlation between two distal protein fragments; points with a positive $\log_2$ fold-change represent correlations that are predicted to increase in strength upon FBP binding. Correlations with a $\log_2$fold-change $\geq 2$ and a false discovery rate $\leq 0.05\%$ (determined from a Wilcoxon ranked-sum test) between PKM2$^{apo}$ and PKM2$^{FBP}$ were designated as AlloHubFs, highlighted in green. A total of 72 AlloHubFs were predicted from this analysis. (D) The positional entropy of the PKM2 fragment-encoded structure correlates linearly with the correlation strength of the fragment. The total mutual information content was computed by summing over the correlations for each of the top AlloHubFs. nMI: normalised mutual information. (E) Left: PKM2 structure depicting the spatial distribution of the top ten predicted AlloHubFs. Right: zoom into a single protomeric chain shown in cartoon representation. AlloHubFs (blue) and FBP (green) are shown as stick models. Black lines indicate minimal distance pathways between the FBP binding pocket and the active site, predicted using Dijkstra's algorithm (see Materials and methods) with the complete set of correlation values as input.

DOI: https://doi.org/10.7554/eLife.45068.023

The following figure supplement is available for figure 5:

**Figure supplement 1.** Solvent accessibility and volume analyses of MD simulations of PKM2.

DOI: https://doi.org/10.7554/eLife.45068.024

MS, we showed that, while FBP and Phe individually have opposing effects on PKM2 oligomerisation, they synergistically stabilise PKM2 tetramers. However, in contrast to PKM2$^{FBP}$ tetramers, PKM2$^{FBP+Phe}$ tetramers have low enzymatic activity. Conversely, the presence of FBP altered the kinetic mechanism of Phe inhibition: Phe alone did not decrease the $k_{cat}$, but instead increased the $K_M^{PEP}$. With FBP present, Phe decreased both the apparent affinity for PEP and the maximal velocity of FBP-bound PKM2.

To identify residues that mediate the interaction between the Phe and FBP allosteric mechanisms, we analysed changes imposed by FBP binding on PKM2 dynamics. MD simulations of PKM2$^{apo}$ and PKM2$^{FBP}$, corroborated by IM-MS data, indicated that the protein does not undergo large conformational changes, in agreement with previously published small-angle X-ray scattering (SAXS) experiments that showed no FBP-driven change in the radius of gyration of PKM2 tetramers (*Yan et al., 2016*). This suggested that ligand-induced conformational changes are likely limited to subtle backbone re-arrangements and side-chain motions. Based on our previous approach (*Pandini et al., 2010*; *Fornili et al., 2013*; *Pandini et al., 2013*), we calculated the mutual information between sampled conformational states from MD trajectories encoded in a coarse-grained representation within the framework of a structural alphabet. Nevertheless, given that proteins have been shown experimentally (*Guerry et al., 2013*; *Salvi et al., 2016*; *Delaforge et al., 2018*; *Kerns et al., 2015*) and computationally (*Markwick et al., 2009*; *Daura et al., 2001*) to sample multiple conformational sub-states with distinct structural properties, explicitly identifying allosteric signals that are representative of the ensemble of protein sub-states is crucial. In order to derive the network of correlated motions from multiple MD trajectories and obtain an ensemble-averaged mutual information network, we developed a new computational approach, AlloHubMat, to identify protein residues as nodes of allosteric interaction networks. AlloHubMat, significantly expands the capabilities of our previous approach, GSATools, which was limited to the comparison of two MD trajectories. AlloHubMat predicts allosteric networks from MD simulations taking into account sampled conformational sub-states and using a consistent numerical framework to measure time-dependent correlated motions, thereby overcoming some of the limitations (*Bürgi et al., 2001*; *Salvi et al., 2016*) of previous approaches. AlloHubMat, enables both the extraction of consensus allosteric networks from

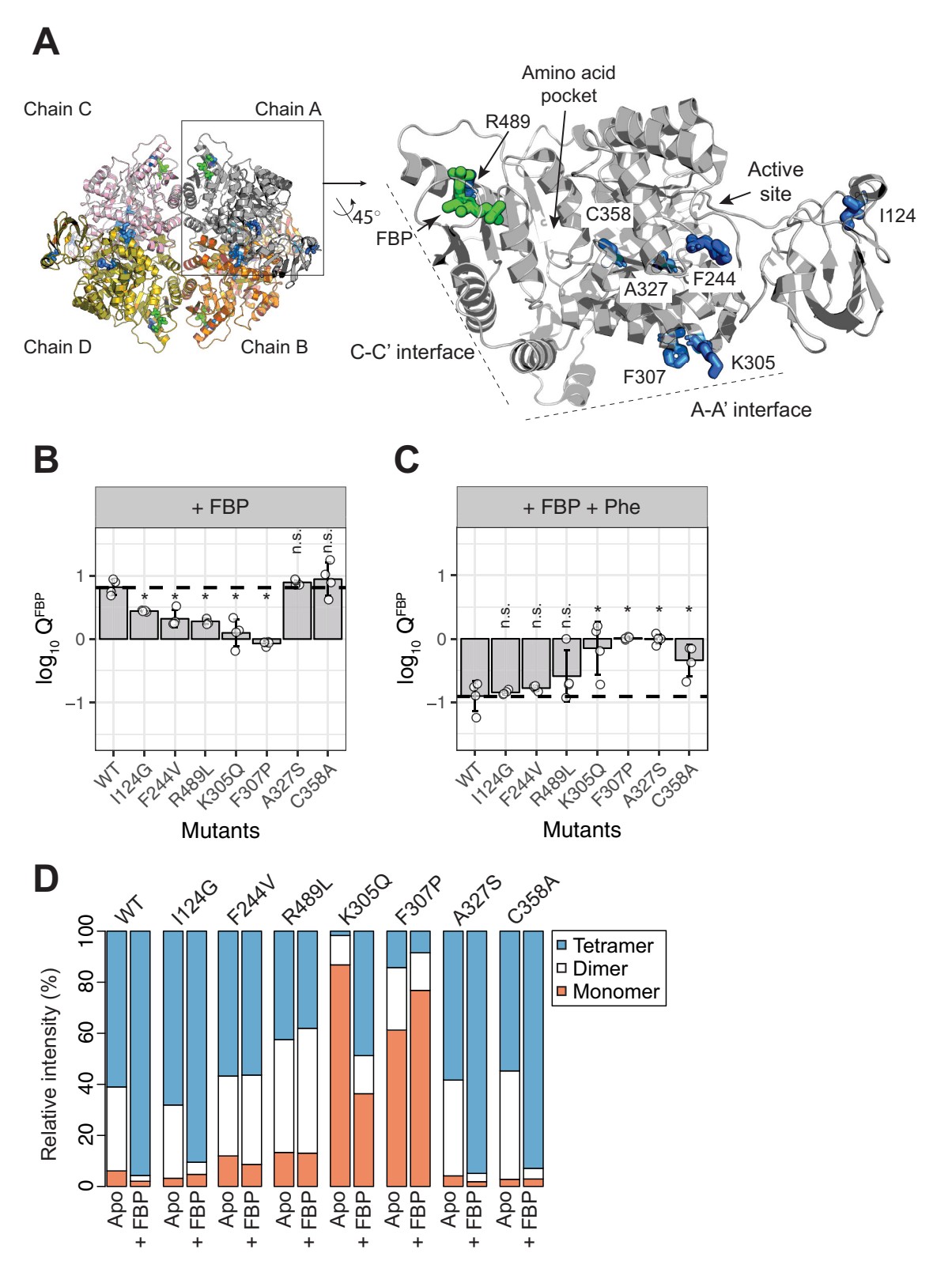

**Figure 6.** AlloHubF mutants (AlloHubMs) either interfere with FBP-induced PKM2 activation or mediate its disruption by Phe. (A) AlloHubF mutants characterised in this study, shown on the PKM2 protomer structure. (B) The allosteric response of PKM2(WT) and AlloHubF mutant enzymatic activities to FBP, quantified by the allosteric coefficient $Q$, which denotes the change of the $K_M^{PEP}$ in the absence and in the presence of saturating concentrations of FBP (see Materials and methods). A $Q$-coefficient > 0, indicates allosteric activation; and $Q$-coefficient < 0 indicates allosteric

*Figure 6 continued on next page*

*Figure 6 continued*

inhibition. The Q-coefficient for PKM2(WT) is shown as a dotted line for comparison. Each of the Q-coefficients of the AlloHubF mutants were statistically compared to PKM2(WT) using a Wilcoxon ranked-sum test ($n = 4$); a p-value<0.05 was deemed significant (denoted by an asterisk); n.s.: not significant. (C) The magnitude of allosteric inhibition by Phe, in the presence of FBP, determined for PKM2(WT) and AlloHubF mutants, quantified by the allosteric co-efficient Q as in (B). (D) Relative abundance of monomers, dimers and tetramers for PKM2 (WT) and PKM2 AlluHubF mutants in the absence or presence of saturating FBP.

DOI: https://doi.org/10.7554/eLife.45068.026

The following figure supplements are available for figure 6:

**Figure supplement 1.** Schematic depicting the integrated computational and experimental strategy used to identify protein residues involved in allosteric regulation.

DOI: https://doi.org/10.7554/eLife.45068.027

**Figure supplement 2.** Sequence conservation analysis of AlloHubFs.

DOI: https://doi.org/10.7554/eLife.45068.028

**Figure supplement 3.** Purified AlloHubF mutants have similar secondary structure content to PKM2(WT).

DOI: https://doi.org/10.7554/eLife.45068.029

**Figure supplement 4.** FBP has a similar affinity for AlloHubMs to that for PKM2(WT), with the exception of PKM2(R489L).

DOI: https://doi.org/10.7554/eLife.45068.030

**Figure supplement 5.** Steady-state enzyme kinetics of the AlloHubMs.

DOI: https://doi.org/10.7554/eLife.45068.031

**Figure supplement 6.** Native mass spectra of the AlloHubMs.

DOI: https://doi.org/10.7554/eLife.45068.032

replicate simulations of a protein in a given liganded state, and also the comparison of such consensus networks to each other.

AlloHubMat revealed candidate residues involved in the allosteric effect of FBP on PKM2 activity. Mutagenesis at several of these positions (I124G, F244V, K305Q, F307P, R489L) disrupted FBP–induced activation demonstrating that AlloHubMat successfully identified *bona fide* mediators of FBP allostery. Some AlloHub mutations also disrupt Phe-induced inhibition suggesting that FBP and Phe elicit allosteric effects through partially overlapping networks of residues. In contrast, PKM2 (F244V) specifically disrupted FBP–induced activation, while maintaining the propensity for allosteric inhibition by Phe. Conversely, mutation of A327 and C358 preserved the ability of FBP to regulate PKM2 but prevented the inhibitory effect of Phe on PKM2$^{FBP}$ enzymatic activity. This finding indicated a role for A327 and C358 in mediating the cross-talk between the allosteric mechanisms elicited by Phe and FBP that allows the former to interfere with the action of the latter. Notably, identification of C358 as an allosteric hub could explain why a chemical modification at this position perturbs PKM2 activity by oxidation (*Anastasiou et al., 2011*). None of the characterised mutants fall within positions 389–429 that differ between PKM2 and the constitutively active PKM1, suggesting that residues that confer differences in the allosteric properties of these two isoforms are dispersed throughout the protein. Our findings highlight the importance of experimentally evaluating the functional role of allosteric residues predicted by computational methods not only in the context of the allosteric effector under investigation but also in response to other potential allosteric effectors that may also be unidentified.

**Table 7.** Apparent steady-state binding constants of FBP to the PKM2 allosteric hub mutants.

| AlloHubM | $K_D^{FBP}$ |
|---|---|
| I124G | (39.5 ± 33.5) nM |
| F244V | (30.7 ± 33.1) nM |
| K305Q | (39.4 ± 34.1) nM |
| F307P | (4.0 ± 12.8) nM |
| A327S | (43.2 ± 48.7) nM |
| C358A | (35.3 ± 23.6) nM |
| R489L | (14.0 ± 2.7) mM |

DOI: https://doi.org/10.7554/eLife.45068.025

**Table 8.** Steady-state Michaelis-Menten kinetic parameters for AlloHubF mutants.

| AlloHubM | Ligand | $K_M^{PEP}$ (mM) | $k_{cat}$ $(s^{-1})$ | $k_{cat}/K_M^{PEP}$ $(s^{-1}$ $mM^{-1})$ |
|---|---|---|---|---|
| I124G | Apo | 1.07 ± 0.13 | 190.27 ± 7.31 | 177.82 ± 56.23 |
| | FBP | 0.29 ± 0.04 | 307.10 ± 6.11 | 1058.97 ± 152.75 |
| | FBP + Phe | 1.44 ± 0.34 | 221.92 ± 17.26 | 153.98 ± 37.88 |
| F244V | Apo | 1.14 ± 0.11 | 237.44 ± 7.36 | 210.44 ± 27.15 |
| | FBP | 0.55 ± 0.07 | 279.62 ± 9.93 | 522.42 ± 87.51 |
| | FBP + Phe | 1.15 ± 0.30 | 222.51 ± 18.29 | 195.87 ± 55.46 |
| K305Q | Apo | 0.01 ± 0.01 | 8.06 ± 0.57 | 790.01 ± 605.56 |
| | FBP | 0.01 ± 0.04 | 8.40 ± 0.43 | 1038.01 ± 535.50 |
| | FBP + Phe | 0.04 ± 0.01 | 12.21 ± 1.40 | 408.37 ± 136.10 |
| F307P | Apo | 0.13 ± 0.01 | 180.47 ± 3.05 | 1413.87 ± 133.12 |
| | FBP | 0.15 ± 0.02 | 227.20 ± 5.02 | 1508.53 ± 190.71 |
| | FBP + Phe | 0.21 ± 0.03 | 328.71 ± 12.37 | 1568.32 ± 268.58 |
| A327S | Apo | 1.37 ± 0.42 | 31.8 ± 2.43 | 23.21 ± 5.79 |
| | FBP | 0.17 ± 0.02 | 119.01 ± 6.86 | 700.06 ± 343.00 |
| | FBP + Phe | 0.15 ± 0.02 | 100.19 ± 5.32 | 667.93 ± 266.00 |
| C358A | Apo | 4.71 ± 0.99 | 214.73 ± 21.40 | 45.59 ± 21.62 |
| | FBP | 0.58 ± 0.18 | 191.13 ± 16.22 | 329.53 ± 90.11 |
| | FBP + Phe | 1.25 ± 0.07 | 199.01 ± 21.87 | 159.20 ± 31.24 |
| R489L | Apo | 0.69 ± 0.19 | 60.05 ± 4.91 | 89.38 ± 36.93 |
| | FBP | 0.36 ± 0.10 | 112.25 ± 7.60 | 317.20 ± 125.71 |
| | FBP + Phe | 1.44 ± 0.40 | 132.99 ± 15.74 | 158.19 ± 122.56 |

DOI: https://doi.org/10.7554/eLife.45068.033

Interestingly, *Zhong et al. (2017)* recently reported that adenosine monophosphate (AMP) and glucose-6-phosphate (G6P) synergistically activate *M. tuberculosis* pyruvate kinase. While the binding of AMP occurs at a pocket equivalent to that of PKM2 for FBP, G6P binds to a different pocket that is also distinct from the equivalent amino acid interaction site on PKM2, indicating an additional allosteric integration mechanism that is similar to the one we describe here. It is therefore tempting to speculate that allosteric synergism upon concurrent binding of different ligands may occur more commonly that previously appreciated.

## AlloHubMs provide insights into the relationship between PKM2 oligomerisation and enzymatic activity

Changes in oligomerisation have been intimately linked to the regulation of PKM2 activity. AlloHubMs A327S and C358A retained both their ability to tetramerise and increase their activity in response to FBP. Furthermore, FBP fails to shift the oligomer equilibrium and does not activate AlloHubMs F307P and R489L. In this context, we found that Phe inhibited the activity of PKM2$^{apo*}$, with concomitant loss of tetramers, however, inhibition of PKM2$^{FBP}$ by Phe occurred within the tetrameric state. The mechanism by which Phe regulates PKM2 oligomerisation is controversial. Our finding that Phe destabilises tetramers is in agreement with previous studies (*Hofmann et al., 1975*), but contrasts with recent reports that Phe stabilises a low activity T-state tetramer (*Morgan et al., 2013*; *Yuan et al., 2018*). Critically, we find that Phe and FBP synergistically promote PKM2 tetramerisation, raising the possibility that the mode of Phe action described in *Morgan et al. (2013)* and *Yuan et al. (2018)* is confounded by the presence of *residual* FBP. It is unclear whether partial FBP occupancy is accounted for in these studies, as the FBP saturation status of PKM2 is not detailed. Co-purification of FBP with recombinant PKM2 has been previously observed (*Morgan et al., 2013*; *Gavriilidou et al., 2018*; *Yan et al., 2016*) and in purifying

recombinant PKM2 for our study, we detected up to more than 0.75 fractional saturation with co-purified FBP. Furthermore, initial attempts, by others, to crystallise Phe-bound PKM2 without FBP required mutation in the protein (R489L) to abrogate FBP binding (*Morgan et al., 2013*), although structures of PKM2(WT) bound to Phe have now been obtained (*Yuan et al., 2018*). Based on our findings, we speculate that the stabilisation of PKM2 tetramers by Phe observed by *Morgan et al. (2013)* and *Yuan et al. (2018)* could be attributed to significant amounts of residually-bound FBP co-purifying from *E. coli*. Consistent with this hypothesis, the small shift in the conformational arrangement ($^{DT}CCS_{He}$) of PKM2$^{FBP}$ tetramers upon addition of Phe (PKM2$^{FBP+Phe}$) may be indicative of subtle conformational changes that reflect a transition from an active R-state to an inactive T-state described before (*Morgan et al., 2013*). In support of this interpretation, the $^{DT}CCS_{He}$ of PKM2$^{FBP+Phe}$ tetramers closely resembles that of PKM2$^{apo*}$.

Therefore, beyond its immediate goals, our study has broader implications for understanding the regulation of PKM2 as it suggests that enzymatic activation can be uncoupled from tetramerisation. This idea resonates well with findings from MD simulations indicating that dynamic coupling between distal sites upon FBP binding can occur in the PKM2 protomer, and suggest that allostery is encoded in the protomer structure (*Yang et al., 2016*; *Naithani et al., 2015*; *Gehrig et al., 2017*). Furthermore, a patient-derived PKM2 mutant (G415R) occurs as a dimer that can bind to FBP but cannot be activated and does not tetramerise (*Yan et al., 2016*). Moreover, SAICAR can activate PKM2(G415R) dimers, in the absence of tetramerisation (*Yan et al., 2016*). The finding that enzymatic activation is not obligatorily linked to tetramerisation is important for studies in intact cells, where distinction between the T- and R- states is not possible and therefore the oligomeric state of PKM2 is frequently used to infer activity (*Anastasiou et al., 2011*; *Christofk et al., 2008a*; *Qi et al., 2017*; *Anastasiou et al., 2012*; *Wang et al., 2017a*; *Wang et al., 2017b*; *Lim et al., 2016*; *Hitosugi et al., 2009*).

## Multiple allosteric inputs in the context of intracellular concentrations of allosteric effectors and other modifications

Our findings also highlight the importance of interpreting allosteric effects detected in vitro in the context of intracellular concentrations of the respective effectors. Reversible binding of FBP to PKM2 in vitro is a well-studied regulatory mechanism. However, our findings reveal that FBP concentrations far exceed the concentration needed for full saturation of PKM2 and, under steady-state cell growth conditions, a significant fraction of PKM2 is already bound to the activator FBP, even in the context of other regulatory cues, such as PTMs, that may influence ligand binding. Furthermore, our results showed that PKM2 inhibition by Phe can occur even under conditions of saturating FBP, both with purified PKM2 and in cell lysates. Taken together, these observations indicate that inhibition by Phe constitutes a physiologically relevant regulatory mechanism that may contribute to maintaining PKM2 at a low activity state, as is often found in cancer cells (*Christofk et al., 2008a*).

Intriguingly, other amino acids can bind the same pocket as Phe, including activators Ser (*Chaneton et al., 2012* ) and His (*Yuan et al., 2018*). Therefore, it is likely that amino acids combined, rather than individually, control PKM2, as also supported by recent findings by *Yuan et al. (2018)*. Further work is warranted to understand how all of these cues are integrated by PKM2.

In summary, our findings reveal that allosteric inputs from distinct ligands are integrated to control the enzymatic activity of PKM2. This is analogous to multiple-input-single-output (MISO) controllers in control system engineering in which multiple transmission signals (allosteric ligands) are integrated to a single receiving signal (enzyme activity) (*Cosentino and Bates, 2011*). It is likely that many proteins can bind to multiple allosteric ligands that co-exist in cells. Whether a systems-control ability in integrating the allosteric effects of multiple ligands with opposing functional signals is a general property of other proteins is not known. Our work does not address the functional consequences of such signal integration in cells. However, identification of mutations, using AlloHubMat, that perturb allosteric responses to specific ligands, alone or in combination, provides an essential means to study both the mechanistic basis of allosteric signal integration as well as the functional consequences of combinatorial allosteric inputs on cellular regulation.

# Materials and methods

## Key resources table

| Reagent type (species) or resource | Designation | Source or reference | Identifiers | Additional information |
|---|---|---|---|---|
| Protein (*Homo sapiens*) | PKM2 | N/A | Uniprot ID: P14618-1 | |
| Cell line (*Homo sapiens*) | HCT116 | ATCC, Manassas, VA, USA | ATCC Cat# CCL-247, RRID:CVCL_0291 | |
| Cell line (*Homo sapiens*) | LN229 | ATCC, Manassas, VA, USA | ATCC Cat# CRL-2611, RRID:CVCL_0393 | |
| Cell line (*Homo sapiens*) | SN12C | Kaelin Lab, DFCI, Boston, MA, USA | RRID:CVCL_1705 | |
| Antibody | Rabbit Anti-PKM2, Clone D78A4 | Cell Signaling Technology | Cat# 4053; RRID:AB_1904096 | 1:1000 in 5% BSA/T-BST |
| Recombinant DNA reagent | pET28a-His-PKM2(WT) | *Anastasiou et al., 2011* | RRID: Addgene_42515 | Available from AddGene (Cambridge MA, USA). See *Supplementary file 2* for PKM2 sequence. |
| Recombinant DNA reagent | pET28a-His-PKM2(I124G) | This study | | 1-step mutagenesis using pET28a-His-PKM2(WT) as template |
| Recombinant DNA reagent | pET28a-His-PKM2(F244V) | This study | | 1-step mutagenesis using pET28a-His-PKM2(WT) as template |
| Recombinant DNA reagent | pET28a-His-PKM2(R489L) | This study | | 1-step mutagenesis using pET28a-His-PKM2(WT) as template |
| Recombinant DNA reagent | pET28a-His-PKM2(K305Q) | This study | | 1-step mutagenesis using pET28a-His-PKM2(WT) as template |
| Recombinant DNA reagent | pET28a-His-PKM2(F307P) | This study | | 1-step mutagenesis using pET28a-His-PKM2 (WT) as template |
| Recombinant DNA reagent | pET28a-His-PKM2(A327S) | This study | | 1-step mutagenesis using pET28a-His-PKM2(WT) as template |
| Recombinant DNA reagent | pET28a-His-PKM2(C358A) | This study | | 1-step mutagenesis using pET28a-His-PKM2(WT) as template |
| Peptide | ITLDNAYMEK [$^{13}C_6^{15}N_2$] | This study | | Synthesised by the Crick Peptide Synthesis STP |
| Peptide | GDLGIEIPAEK [$^{13}C_6^{15}N_2$] | This study | | Synthesised by the Crick Peptide Synthesis STP |
| Peptide | APIIAVTR[$^{13}C_6^{15}N_4$] | This study | | Synthesised by the Crick Peptide Synthesis STP |
| Peptide | LFEELVR[$^{13}C_6^{15}N_4$] | This study | | Synthesised by the Crick Peptide Synthesis STP |
| Peptide | LAPITSDPTEATA VGAVEASFK[$^{13}C_6^{15}N_2$] | This study | | Synthesised by the Crick Peptide Synthesis STP |
| Chemical compound | Potassium phospho enolpyruvate (2,3-$^{13}C_2$) | Cambridge Isotope Laboratories | CLM-3398-PK | |

*Continued on next page*

*Continued*

| Reagent type (species) or resource | Designation | Source or reference | Identifiers | Additional information |
|---|---|---|---|---|
| Chemical compound | D-fructose 1,6-bisphosphate sodium salt hydrate (U-$^{13}C_6$) | Cambridge Isotope Laboratories | CLM-8962-PK | |
| Chemical compound | L-Phe (ring-$^{13}C_6$) | Cambridge Isotope Laboratories | CLM-1055-PK | |
| Chemical compound | L-Ser ($^{13}C_6$) | Cambridge Isotope Laboratories | CLM-1574-H | |
| Software | sa_encode.R | This study | | Software to encode trajectory into stacked alignment of structural alphabet strings (*Kleinjung, 2019*; copy archived at https://github.com/elifesciences-publications/ALLOHUBMAT) |
| Software | kabsch.R | This study | | Kabsch superpositioning routine (*Kleinjung, 2019*; copy archived at https://github.com/elifesciences-publications/ALLOHUBMAT) |
| Software | MI.R | This study | | Mutual Information and other entropy metrics between two character vectors, here intended for two alignment columns (*Kleinjung, 2019*; copy archived at https://github.com/elifesciences-publications/ALLOHUBMAT) |
| Software | Xcalibur QualBrowser | Thermo Fisher Scientific (Waltham MA, USA) | N/A | |
| Software | Tracefinder v4.1 | Thermo Fisher Scientific (Waltham MA, USA) | N/A | |

## Metabolite analyses by liquid chromatography-mass spectrometry (LC-MS)

The LC-MS method was adapted from *Zhang et al. (2012)*. Briefly, samples were injected into a Dionex UltiMate LC system (Thermo Scientific; Waltham MA, USA) with a ZIC-pHILIC (150 mm x 4.6 mm, 5 µm particle) column (Merck Sequant; MilliporeSigma, Burlington MA, USA). A 15 min elution gradient of 80% Solvent A to 20% Solvent B was used, followed by a 5 min wash of 95:5 Solvent A to Solvent B and 5 min re-equilibration, where Solvent B was acetonitrile (Optima HPLC grade; Sigma Aldrich, St. Louis MS, USA) and Solvent A was 20 mM ammonium carbonate in water (Optima HPLC grade; Sigma Aldrich, St. Louis MS, USA). Other parameters were used as follows: injection volume 10 µL; autosampler temperature 4℃; flow rate 300 µL/min; column temperature 25℃. MS was performed using positive/negative polarity switching using an Q Exactive Orbitrap (Thermo Scientific; Waltham MA, USA) with a HESI II (Heated electrospray ionization) probe. MS parameters were used as follows: spray voltage 3.5 kV and 3.2 kV for positive and negative modes, respectively;

probe temperature 320°C; sheath and auxiliary gases were 30 and 5 arbitrary units, respectively; full scan range: 70 m/z to 1050 m/z with settings of AGC target ($3 \times 10^6$) and mass resolution as Balanced and High (70,000). Data were recorded using Xcalibur 3.0.63 software (Thermo Scientific; Waltham MA, USA). Before analysis, Thermo Scientific Calmix solution was used as a standard to perform mass calibration in both ESI polarities and ubiquitous low-mass contaminants were used to apply lock-mass correction to each analytical run in order to enhance calibration stability. Parallel reaction monitoring (PRM) acquisition parameters: resolution 17,500, auto gain control target $2 \times 10^5$, maximum isolation time 100 ms, isolation window m/z 0.4; collision energies were set individually in HCD (high-energy collisional dissociation) mode. Quality control samples were generated by taking equal volumes of each sample and pooling them, and subsequently analysing this mix throughout the run to assess the stability and performance of the system. Qualitative and quantitative analysis was performed using Xcalibur Qual Browser and Tracefinder 4.1 software (Thermo Scientific; Waltham MA, USA) according to the manufacturer's workflows.

## Cell lines and cell culture

HCT116 and LN229 were obtained from the American Type Culture Collection (ATCC, Manassas, VA, USA). SN12C were a gift from William Kaelin (Harvard, Boston, USA). All cell lines were cultured in RPMI 1640 medium (Gibco, 31840) supplemented with 10% foetal calf serum (FCS), 2 mM glutamine, 100 U/mL penicillin/streptomycin in a humidified incubator at 37°C, 5% $CO_2$. All cell lines were tested mycoplasma-free and cell identity was confirmed by short tandem repeat (STR) profiling by The Francis Crick Institute Cell Services Science Technology Platform.

## Metabolite extraction and cell volume calculations

24 hr prior to the experiment, cells were seeded in 6 cm dishes in RPMI media containing 10% dialysed FCS (3500 Da MWCO, PBS used for dialysis). An hour prior to treatment, the media were refreshed, and then changed again at t = 0 to RPMI with or without 11 mM glucose; or to HBSS (H2969-500mL; Sigma Aldrich, St. Louis MS, USA) with or without supplemented amino acids as described in the text. Four technical replicate plates were used for each condition, and 2–3 plates for each cell line were used to count cells and measure mean cell diameter which was then used to determine cell volume in order to estimate intracellular concentrations. After 1 hr of treatment, plates were washed twice with ice-cold PBS, and 725 µl of dry-ice-cold methanol was used to quench the cells. Plates were scraped and contents were transferred to Eppendorf tubes on ice containing 180 µl $H_2O$ and 160 µl $CHCl_3$. A further 725 µL methanol was used to scrape each plate and added to the same Eppendorf. Samples were vortexed and sonicated in a cold sonicating water bath 3 times for 8 mins each time. Extraction of metabolites was allowed to proceed at 4°C overnight, before spinning down precipitated material and then drying down supernatant. To split polar and apolar phases, dried metabolites were resuspended in a 1:3:3 mix of $CHCl_3$/MeOH/$H_2O$ (total volume of 350 µl). Polar metabolites in the aqueous phase were then analysed by LC-MS. To enable absolute quantification of metabolites of interest, known quantities of $^{13}C$-labelled versions of those metabolites were added into lysates, all purchased from Cambridge Isotopes (Tewksbury MA, USA). Previously determined cell numbers and volumes were then used to determine intracellular concentrations.

| Metabolite | Formula | Exact mass | Pos. mode M/z | Neg. mode M/z | Mode used | RT (min) |
|---|---|---|---|---|---|---|
| FBP | $C_6H_{14}O_{12}P_2$ | 339.99611 | 341.00394 | 338.98829 | neg. | 13.42 |
| $^{13}C_6$-FBP | $^{13}C_6H_{14}O_{12}P_2$ | 346.01621 | 347.02404 | 345.00839 | neg. | 13.42 |
| PEP | $C_3H_5O_6P$ | 167.98241 | 168.99023 | 166.97458 | neg. | 13.58 |
| $^{13}C_2$-PEP | $^{13}C_2C_1H_5O_6P$ | 169.98911 | 170.99693 | 168.98128 | neg. | 13.58 |
| Ser | $C_3H_7NO_3$ | 105.0426 | 106.05043 | 104.03478 | pos. | 13.14 |
| $^{13}C_3$-Ser | $^{13}C_3H_7NO_3$ | 108.05265 | 109.06048 | 107.04483 | pos. | 13.14 |
| Phe | $C_9H_{11}NO_2$ | 165.07899 | 166.08681 | 164.07116 | pos. | 9.7 |
| $^{13}C_6$-Phe | $^{13}C_63H_{11}NO_2$ | 171.09909 | 172.10691 | 170.09126 | pos. | 9.7 |

## Targeted proteomics

*Trypsin digestion* – Cell extracts containing 50 μg of total protein were precipitated by adding six volumes of ice-cold acetone (pre-cooled to −20°C). The samples were allowed to precipitate overnight at −20°C and centrifuged at 8000 g, for 10 min at 4°C, to collect the pellet. The supernatant was carefully decanted and the residual acetone was evaporated at ambient temperature. The pellet was dissolved in 50 mM TEAB, reduced with 10 mM DTT and alkylated with 20 mM iodoacetamide. After alkylation, the proteins were digested with 1 μg of trypsin overnight at 37°C. After digestion, each sample was spiked with a mixture of five heavy labelled standards. For MS analysis, 1 μg of peptides were loaded onto 50 cm Easy Spray C18 column (Thermo Scientific; Waltham MA, USA).

*Analysis of peptides by LC-tandem MS (LC-MS/MS)* – Mass spectrometric analysis was performed using a Dionex U3000 system (SRD3400 degasser, WPS-3000TPL-RS autosampler, 3500RS nano pump) coupled to a QExactive electrospray ionisation hybrid quadruole-orbitrap mass spectrometer (Thermo Scientific; Waltham MA, USA). Reverse phase chromatography was performed with a binary buffer system at a flow rate of 250 nL/min. Mobile phase A was 5% DMSO in 0.1% formic acid and mobile phase B was 5% DMSO, 80% acetonitrile in 0.1% formic acid. The digested samples were run on a linear gradient of solvent B (2–35%) in 90 min, the total run time including column conditioning was 120 min. The nanoLC was coupled to a QExactive mass spectrometer using an EasySpray nano source (Thermo Scientific; Waltham MA, USA). The spray conditions were: spray voltage + 2.1 kV, capillary temperature 250°C and S-lens RF level of 55. For the PRM (parallel reaction monitoring) experiments, the QExactive was operated in data independent mode. A full scan MS1 was measured at 70,000 resolution (AGC target $3 \times 10^6$, 50 ms maximum injection time, m/z 300–1800). This was followed by ten PRM scans triggered by an inclusion list (17,500 resolution, AGC target $2 \times 10^5$, 55 msec maximum injection time). Ion activation/dissociation was performed using HCD at normalised collision energy of 28.

*Data analysis of PRM* – Peptides to be targeted in the PRM-MS analysis were selected previously by analysing trypsin digested cell extracts from the three cell lines of interest. Peptides providing a good signal and identification score representing the two PKM isoforms (PKM1/2) were selected for the analysis. The corresponding heavy isotope-labelled standards were synthesised in-house. The PRM method was developed for the QExactive using Skyline 4.1.0.18169. The heavy labelled peptide standards were used to create the precursor (inclusion) list. When measuring the abundance of PKM1/2 in the cell extracts signal extraction was performed on + 2 precursor ions for both heavy and light forms of the peptides. A peptide was considered identified if at least four overlapping transitions were detected. Quantitation was performed using MS2 XICs where the top three transitions were summed and used for quantitation. Data processing was performed with Skyline which was used to generate peak areas for both light and heavy peptides. Extracted ion chromatograms were visually inspected and peak boundaries were corrected and potential interferences removed. The data was subsequently exported in Excel to calculate absolute quantities of the 'native' peptides and to determine reproducibility (CV %) of the measurements. See *Supplementary file 1*. Data are available via ProteomeXchange with identifier PXD010334.

## Recombinant protein expression and purification

Allosteric hub mutant plasmids were generated through a single-step PCR reaction using hot-start KOD polymerase (Merck Millipore; Burlington MA, USA) and a pET28a-His-PKM2(WT) template plasmid (# 42515 AddGene; Cambridge MA, USA). Plasmids were sequence-verified by Sanger Sequencing (Source Bioscience; Nottingham, UK). 40 ng of pET28a-His-PKM2 (wild-type or mutant) was transformed into 50 μL *E. coli* BL21(DE3)pLysS (60413; Lucigen, Middleton WI, USA). Colonies were inoculated in LB medium at 37°C and grown to an optical density of 0.8 AU (600 nm), at which point expression of the N-terminal His$_6$-PKM2(WT) was induced with 0.5 mM isopropyl β-D-1 thiolgalacto-pyranoside (Sigma Aldrich, St. Louis MS, USA). The culture was grown at 24°C for 16–18 hr. Cells were harvested by centrifugation and the pellet was re-suspended in cell lysis buffer (50 mM Tris-HCl pH 7.5, 10 mM MgCl$_2$, 200 mM NaCl, 100 mM KCl and 10 mM imidazole) supplemented with the EDTA-free Complete protease inhibitor cocktail (Sigma Aldrich, St. Louis MS, USA). Cells were lysed by sonication at 4°C. DNase was added at 1 μL/mL before centrifugation of the lysate at 20000 g for 1 hr at 4°C. The supernatant (the water-soluble cell fraction) was loaded onto a HisTrap HP nickel-charged IMAC column (GE; Boston MA, USA) and was washed with five column-volumes of

wash buffer [10 mM HEPES pH 7.5, 10 mM MgCl$_2$, 100 mM KCl, 10 mM imidazole and 0.5 mM tris-2-carboxyethyl phosphine hydrochloride (TCEP; Sigma Aldrich, St. Louis MS, USA)]. After consecutive wash steps, the protein was eluted from the IMAC column with elution buffer buffer (10 mM HEPES pH 7.5, 10 mM MgCl$_2$, 100 mM KCl, 250 mM imidazole and 0.5 mM TCEP). The N-terminal His$_6$-epipope tag was cleaved with at 4˚C for ~ 18 hr in cleavage buffer (50 mM Tris-HCl pH 8.0, 10 mM CaCl$_2$) with recombinant bovine thrombin (see *Supplementary file 2*), immobilised on agarose beads. Purified recombinant PKM2 was eluted from the thrombin-agarose column. Affinity purification was followed by size-exclusion chromatography on a HiLoad 16/60 Superdex 200 pg column (28-9893-35; GE, Boston MA, USA) at 500 mL/min flow rate with protein storage buffer (10 mM HEPES pH 7.5, 10 mM MgCl$_2$, 100 mM KCl, and 0.5 mM TCEP) at 4˚C. Eluted PKM2 was collected and concentrated to a final protein concentration of ~7 mg/mL with centrifugal concentrating filters (Vivaspin 20, 10 kDa molecular-weight cut-off, 28-9323-60; GE, Boston MA, USA). Protein purity was assessed by SDS-PAGE. The final concentration of the protein was obtained by measuring the fluorescence absorbance spectrum between 240 nm and 450 nm. The concentration was calculated using a molar extinction coefficient of 29,910 M$^{-1}$ cm$^{-1}$ at 280 nm.

## Quantification of residual D-fructose 1,6-bisphosphate co-purified with recombinant PKM2

Molar amounts of D-fructose 1,6-bisphosphate (FBP) that co-purified with recombinant PKM2 were measured using an aldolase enzyme assay that comprises three coupled enzymatic steps (*Figure 2—figure supplement 1A*). The reaction mixture contained 20 mM Tris-HCl pH 7.0, 50 μM NADH, 0.7 U/mL glycerol 3-phosphate dehydrogenase (G-3-PDH), 7 U/ml triose phosphate isomerase (TPI) and the supernatant of 5–50 μM purified recombinant PKM2 after heat-precipitation at 90˚C. G-3-PDH and TPI from rabbit muscle were purchased as a mixture (50017, Sigma Aldrich; St. Louis MS, USA). The reaction was initiated by adding between 0.008–0.016 U/ml rabbit muscle aldolase (A2714, Sigma Aldrich; St. Louis MS, USA) to a total reaction volume of 100 μL. Two molecules of NADH are oxidised for each molecule of FBP consumed. NADH oxidation was monitored over time at 25˚C in a 1 mL quartz cuvette (1 cm path-length) by measuring the NADH absorption signal at 340 nm using a Jasco V-550 UV-Vis spectrophotometer. For the assay calibration, known amounts of FBP from a powder stock were used instead of the heat-precipitated PKM2 supernatant.

## Measurement of PKM2 steady-state enzyme kinetics

Steady-state enzyme kinetic measurements of PKM2 were performed using a Tecan Infinite 200-Pro plate reader (Tecan, Männedorf Zürich, Switzerland). Initial velocities for the forward reaction (phosphoenolpyruvate and adenosine diphosphate conversion to pyruvate and adenosine triphosphate) were measured using a coupled reaction with rabbit muscle lactate dehydrogenase (Sigma Aldrich, St. Louis MS, USA). The reaction monitored the oxidation of NADH ($\varepsilon_{340\ nm} = 6220\ M^{-1}cm^{-1}$) at 37 ˚C in a buffer containing 10 mM Tris-HCl pH 7.5, 100 mM KCl, 5 mM MgCl$_2$ and 0.5 mM TCEP. Initial velocity versus substrate concentrations for phosphoenolpyruvate were measured in the absence and in the presence of allosteric ligands, in a reaction buffer containing 180 μM NADH and 8 U lactate dehydrogenase. Reactions were initiated by adding phosphenolpyruvate (PEP) at a desired concentration, with adenosine diphosphate (ADP) at a constant concentration of 5 mM. A total protein concentration of 5 nM PKM2 was used for all enzyme reactions, in a total reaction volume of 100 μL per well. Kinetic constants were determined by fitting initial velocity curves to Michaelis-Menten steady-state kinetic models.

## Measurements of FBP binding to PKM2

The affinity of PKM2 for FBP was measured by titrating small aliquots of a concentrated stock solution of FBP into 5 μM recombinant PKM2 and recording intrinsic fluorescence emission spectra of PKM2. Spectra were recorded using a Jasco FP-8500 spectrofluorometer with an excitation wavelength of 280 nm (bandwidth of 2 nm) and emission scanned from 290 nm to 450 nm (bandwidth of 5 nm) in a 0.3 cm path length quartz cuvette (Hellma Analytics; Muellheim, Germany) at 20˚C. The ratio of the emission intensities at 325 and 350 nm was plotted against the concentration of the titrant. Binding curves were fit to a model assuming a 1:1 binding stoichiometry (1 FBP molecule per monomer of PKM2) with a non-linear least squares regression fit of the following equation:

$$S_{obs} = \frac{S_{P1}[P] + S_{PL1}[PL]}{S_{P2}[P] + S_{PL2}[PL]}$$

with $[P]=[P_0]-[PL]$ and $[PL]$ calculated using

$$[PL] = \left( \frac{K_D + [P_0] + [L_0] - \sqrt{(K_D + [P_0] + [L_0])^2 - 4[P_0][L_0]}}{2} \right)$$

where the spectral signal $S_{obs}$ is the ratio of fluorescence emissions at wavelengths 1 (325 nm) and 2 (350 nm); $S_{P1}$, $S_{P2}$, $S_{PL1}$ and $S_{PL2}$ are the fluorescence extinction coefficients of the free protein $P$ and the protein-ligand complex $PL$ at wavelengths 1 and 2, respectively; $[P_0]$ and $[L_0]$ are total concentrations of protein and ligand, respectively; and $K_D$ is the apparent dissociation constant. The total free protein concentration ($[P_0]$) was corrected by subtracting the percentage of protein pre-bound to co-purified FBP, as determined from the aldolase assay.

Ideally, the concentrations of protein and ligand used for binding experiments should be in the same range as the measured $K_D$. Microscale thermophoresis (MST, see below), which would offer sufficient sensitivity to use lower PKM2 concentrations, did not reveal a change in the thermophoretic properties of PKM2 upon FBP binding. Owing to the low quantum yield of intrinsic PKM2 fluorescence, measurement of FBP binding to PKM2 by fluorometry, necessitated the use of high PKM2 concentrations relative to the $K_D^{FBP}$, in order to obtain measurements with acceptable signal-to-noise ratios. However, when $[P_0] >> K_D$, the solution of the above equation for $K_D$ is associated with a high numerical error. In order to use the estimated $K_D^{FBP}$, despite this limitation, to calculate the fractional occupancy of PKM2 by FBP in cells, we derived the average $K_D^{FBP}$ value from ten independent replicate FBP titration measurements and propagated the errors from these measurements using the equation:

$$\sigma K_D = K_D \sqrt{\sum_{i=1}^{n} \left( \frac{\sigma K_D^i}{K_D^i} \right)^2}$$

The propagated error was then added to the average $K_D^{FBP}$ to obtain an upper limit estimate. The $K_D^{FBP}$ shown in *Table 1* was determined in a separate experiment.

## Measurements of phenylalanine (Phe) and serine (Ser) binding to PKM2

The binding of Phe and Ser to PKM2 was measured using microscale thermophoresis (MST) on a Monolith NT.115 instrument (Nanotemper Technologies; Munich, Germany). First, PKM2 was fluorescently labelled with an Atto-647 fluorescein dye (NT-647-NHS; Nanotemper Technologies; Munich, Germany). 250 μL of 20 μM recombinant PKM2 was labelled with 250 μL of 60 μM dye in a buffer containing 100 mM bicarbonate pH 8.5% and 50% DMSO for 30 min at room temperature in the dark. Free dye was separated from labelled PKM2 using a NAP-5 20 ST size-exclusion column (GE; Boston MA, USA). Labelled PKM2, at a constant concentration of 30 nM, was titrated with either Phe (up to 5 mM) or Ser (up to 10 mM) in a buffer containing 10 mM HEPES pH 7.5, 100 mM KCl, 5 mM MgCl$_2$, 0.5 mM TCEP and 0.1% tween-20. Prior to each thermophoresis measurement, capillary scans were obtained to determine sample homogeneity. Binding curves were fit assuming a 1:1 stoichiometry.

## Analysis of the steady-state kinetics of PKM2 enzyme activity inhibition by Phe

In order to assign the mechanism with which Phe inhibition of PKM2 occurs, in the absence and in the presence of FBP, the dependence of the enzyme kinetic constants $K_M$, $k_{cat}$ and $\frac{k_{cat}}{K_M}$ on the concentration of Phe were determined. Steady-state measurements of PKM2 enzyme activity (as described above) were performed by titrating the substrate PEP at several different concentrations of Phe and a constant concentration of 5 mM ADP. In order to investigate the allosteric K-type effect of Phe on enzyme affinity for its substrate PEP, a single-substrate-single-effector paradigm was assumed.

Under equilibrium conditions the rate equation of the general modifier mechanism reveals apparent values of $K_M$ and $k_{cat}$:

$$\frac{v}{[E]_t} = \frac{k_2 \frac{1 + \beta \frac{[x]}{\alpha K_x}}{1 + \frac{[x]}{\alpha K_x}}}{K_M \frac{1 + \frac{[x]}{K_x}}{1 + \frac{[x]}{\alpha K_x} + [S]}}$$

$$\frac{v}{[E]_t} = \frac{k_{cat}^{app} [S]}{K_M^{app} + [S]}$$

where $[E]_t$ is the concentration of enzyme active sites, $x$ is Phe, $S$ is the substrate (phosphoenolpyruvate), $K_x$ is the dissociation constant of the specific component of the enzyme mechanism, $\alpha$ is the reciprocal allosteric coupling constant and $\beta$ is the factor by which the inhibitor affects the catalytic rate constant $k_2$ (**Baici, 2015**). The kinetic constants $K_M^{app}$, $k_{cat}^{app}$ and $\left(\frac{k_{cat}}{K_M}\right)^{app}$ on the concentration of the modifier (X; Phe) can then be written as follows (**Baici, 2015**):

$$k_{cat}^{app} = k_2 \frac{1 + \beta \frac{[X]}{\alpha K_x}}{1 + \frac{[X]}{\alpha K_x}}$$

$$K_M^{app} = K_S \frac{1 + \frac{[X]}{K_x}}{1 + \frac{[X]}{\alpha K_x}}$$

$$\left(\frac{k_{cat}}{K_M}\right)^{app} = \frac{k_2}{K_S} \frac{1 + \beta \frac{[X]}{\alpha K_x}}{1 + \frac{[X]}{K_x}}$$

Rate curves measuring the dependence of [Phe] on the three kinetic constants were fit to the equations above to solve for the constants $K_S$, $K_x$, $\alpha$ and $\beta$. The kinetic mechanism was assigned based on the topology of rate-modifier mechanisms detailed by **Baici (2015)**.

## Native Mass Spectrometry

PKM2 samples were buffer exchanged into 200 mM ammonium acetate (Fisher Scientific; Loughborough, UK) using Micro Bio-Spin six chromatography columns (Bio-Rad Laboratories; Hercules CA, US). Samples were diluted to a final protein concentration of between 5 µM and 20 µM depending on the experiment. Ligands were dissolved in 200 mM ammonium acetate and added to the protein prior to MS analysis.

Native mass spectrometry experiments were performed across three different instruments: a Ultima Global (Micromass; Wilmslow, UK) extended for high mass range, a modified Synapt G2 (Waters Corp, Wilmslow, UK) where the triwave assembly was replaced with a linear drift tube, and a Synapt G2-Si. Samples were analysed in positive ionization mode using nano-electrospray ionization, in which the sample is placed inside a borosilicate glass capillary (World Precision Instruments; Stevenage, UK) pulled in-house on a Flaming/Brown P-1000 micropipette puller (Sutter Instrument Company; Novato CA, USA) and a platinum wire is inserted into the solution to allow the application of a positive voltage. All voltages used were kept as low as possible to achieve spray while keeping the protein in a native-like state. Typical conditions used were capillary voltage of ~ 1.2 kV, cone voltage of ~ 10 V and a source temperature of 40°C.

## Ion mobility mass spectrometry (IM-MS)

IM-MS measurements were performed on an in-house modified Synapt G2 in which the original tri-wave assembly was replaced with a linear drift tube with a length of 25.05 cm. Drift times were measured in helium at a temperature of 298.15 K and pressure of 1.99–2.00 torr. Conditions were kept constant across each run. Measurements were performed at least twice for each sample and averaged. Mobilities for all charge states were converted into rotationally averaged collision cross sections ($^{DT}CCS_{He}$) using the Mason-Schamp equation (**Revercomb and Mason, 1975**), and further

converted into a single global collision cross section distribution per species which all charge states contribute towards in proportion to their intensity in the mass spectrum (*Pacholarz et al., 2014*).

## Surface-Induced dissociation (SID)

SID was performed on a prototype instrument at Waters Corp (Wilmslow, UK). Briefly, sample was ionized via nano-ESI and the species of interest was mass selected in the quadrupole. Ions were then accelerated towards the gold surface of an SID device and underwent a single, high energy collision. Fragments were refocused and mass analysed.

## Molecular dynamics simulations in explicit solvent

Molecular dynamics (MD) simulations of tetrameric human PKM2 were performed with the GRO-MACS 5.2 engine (*Hess et al., 2008*), using the Gromos 53a6 force field parameter sets (*Schmid et al., 2011*) and SPC-E water molecules. The input coordinates for PKM2$^{apo}$ were extracted from the Protein Data Bank crystal structure 3bjt (*Christofk et al., 2008a*) and coordinates for PKM2$^{FBP}$ were extracted from the crystal structure 3u2z (*Anastasiou et al., 2012*). The force-field parameters for FBP were determined using a quantum mechanical assignment of the partial charges using the ATB server (*Malde et al., 2011*). Structures were prepared as previously described (*Gehrig et al., 2017*). Briefly, structures were solvated in a dodecahedral period box, such that the distance between any protein atom and the periodic boundary was a minimum of 1.0 nm. The system charge was neutralised by adding counter ions to the solvent (Na$^+$ and Cl$^-$). Equations of motion were integrated using the leap-frog algorithm (*Berendsen et al., 1984*) with a two fs time step. The system was equilibrated for five ns in the NVT ensemble at 300 K and 1 bar. This was followed by a further five ns equilibration in the NPT ensemble. Following equilibration, five replicate production run simulations were performed for 400 ns under constant pressure and temperature conditions, 1 bar and 300 K. Temperature was regulated using the velocity-rescaling algorithm (*Bussi et al., 2007*), with a coupling constant of 0.1. Covalent bonds and water molecules were restrained with the LINCS and SETTLE methods, respectively. Electrostatics were calculated with the particle mesh Ewald method, with a 1.4 nm cut-off, a 0.12 nm FFT grid spacing and a four-order interpolation polynomial for the reciprocal space sums.

## Prediction of allosteric hub residues with AlloHubMat

MD trajectories were coarse-grained with the M32K25 structural alphabet using the GSAtools (*Pandini et al., 2013*). From the fragment-encoded trajectory, the mutual information between each combination of fragment positions $\langle I^n(C_i;\ C_j) \rangle$ was determined for multiple replicate 400 ns trajectories for PKM2$^{apo}$ and PKM2$^{FBP}$. Each trajectory was sub-divided into 20 non-overlapping blocks with an equal time length of 20 ns each. For each trajectory block ($B$), correlated conformational motions for all pairs of fragments ($i$, $j$) were calculated as the normalised mutual information between each fragment pair in the fragment-encoded alignment $I^n_B(C_i;\ C_j)$:

$$I^n_B(C_i;\ C_j) = \frac{I_B(C_i;\ C_j) - \varepsilon_B(C_i;\ C_j)}{H_B(C_i,\ C_j)}$$

where the columns of the structural fragment alignment are given by $C_i$ and $C_j$, $I_B(C_i;\ C_j)$ is the mutual information, $\varepsilon_B(C_i;\ C_j)$ is the expected finite size error and $H_B(C_i,\ C_j)$ is the joint entropy (*Pandini et al., 2012*). In this framework, the mutual information is given by:

$$I_B(C_i;\ C_j) = \sum \sum p(c_i,\ c_j) \log \frac{p(c_i,\ c_j)}{p(c_i)\,p(c_i)}$$

where the two columns in the structural alphabet alignment $C_i$ and $C_j$ are random variables with a joint probability mass function $p(c_i,\ c_j)$, and marginal probability mass functions $p(c_i)$ and $p(c_j)$. The joint entropy $H(C_i;\ C_j)$ is defined as:

$$H_B(C_i,\ C_j) = -\sum \sum p(c_i,\ c_j) \log p(c_i,\ c_j)$$

The discrete mutual information calculated for finite state probabilities can be significantly

affected by random and systematic errors. In order to account for this, we subtract an error term $\varepsilon_B\left(C_i;\ C_j\right)$ given by:

$$\varepsilon_B\left(C_i,\ C_j\right) = \frac{B^*_{C_i,\ C_j} - B^*_{C_i} - B^*_{C_j} + 1}{2N}$$

where $N$ is the sample size and $B^*_{C_i,\ C_j}$, $B^*_{C_i}$ and $B^*_{C_j}$ are the number of states with non-zero probabilities for $C_iC_j$, $C_i$ and $C_j$, respectively (*Roulston, 1999*).

With the goal of identifying conformational sub-states and their respective probabilities from each of the trajectories, eigenvalue decomposition was used to compute the geometric evolution of the protein backbone correlations (given by the mutual information between structural-alphabet fragments). The elements of the mutual information matrix are proportional to the square of the displacement, so the square root of the matrix is required to examine the extent of the matrix overlap:

$$d(A,B) = \sqrt{tr\left[\left(A^{\frac{1}{2}} - B^{\frac{1}{2}}\right)^2\right]}$$

$$= \sqrt{tr\left[A + B - 2A^{\frac{1}{2}}B^{\frac{1}{2}}\right]}$$

$$= \left[\sum_{i=1}^{3N}\left(\lambda_i^A + \lambda_i^B\right) - 2\sum_{i=1}^{3N}\sum_{j=1}^{3N}\left(\lambda_i^A + \lambda_i^B\right)^{\frac{1}{2}}\left(v_i^A + v_i^B\right)^2\right]^{\frac{1}{2}}$$

$$\Omega_{A;B} = 1 - \frac{d(A,B)}{\sqrt{trA + trB}}$$

where $\lambda_i^A$ and $\lambda_i^B$ denote the eigenvalues and $v_i^A$ and $v_i^B$ the eigenvectors of mutual information matrices $A$ and $B$, $N$ is the number of structural alphabet fragments. The covariance matrix overlap ($\Omega_{A;B}$) ranges between 0 when matrices $A$ and $B$ are orthogonal, and 1 when they are identical.

Using the approach detailed above, the covariance overlap ($\Omega_{A;B}$) was determined from time-contiguous mutual information matrices extracted from non-overlapping trajectory blocks. Conformational sub-states were identified as containing a high degree of similarity between time-contiguous mutual information matrices. Using this approach, an ensemble-averaged mutual information matrix for PKM2apo and for PKM2FBP was determined by averaging over all mutual information matrices identified from each conformational sub-state of the multiple replicate simulations of both liganded states.

The above approach made it possible to subtract the mutual information matrices of correlated motions of the holo- from the apo-state. A difference mutual information matrix was constructed by subtracting the ensemble-averaged matrix of PKM2FBP from PKM2apo. Allosteric hub fragments (AlloHubFs) were identified from this difference mutual information matrix, as those fragments with the highest log2-fold change in the coupling strength and a p-value associated with the change of less than 0.01.

## Estimation of the configurational entropy from explicit solvent MD simulations

The configurational entropy of MD trajectories of PKM2apo and PKM2FBP were estimated using a formulism proposed by Schlitter (*Schlitter, 1993*). For a classical-mechanical system the configurational entropy is given by:

$$S' = \frac{1}{k_BT}\ln\left(1 + \frac{k_BTe^2}{\hbar2}M\sigma_{ij}\right)$$

where

$$\hbar = \frac{h}{2\pi}$$

$k_B$ is the Boltzmann constant, T is temperature, e is Euler's number, $h$ is Plank's constant and $M\sigma_{ij}$ is the mass-weighted covariance matrix of the form:

$$\sigma_{ij} = \left\langle \left(x_i - \langle x_i \rangle\right)\left(x_j - \langle x_j \rangle\right)\right\rangle$$

where $x_i$ and $x_j$ are positional coordinates of the MD trajectory in Cartesian space.

## Theoretical collision cross section calculations

Structural models of the 15 + monomer, 23 + A-A' and C-C' dimers and 33 + tetramers were generated from the PDB crystal structure 3bjt (*Christofk et al., 2008b*) and simulated *in vacuo* using the OPLS-AA/L force-field parameter set (*Kaminski et al., 2001*). Systems were minimised using the Steepest Descent algorithm for $5 \times 10^6$ steps, with a step size of 1 J mol$^{-1}$ nm$^{-1}$ and a maximal force tolerance of 100 kJ mol$^{-1}$ nm$^{-1}$. Next, systems were equilibrated at consecutively increasing temperatures (100 K, 200 K and 300 K) each for five ns, with the Berendsen temperature coupling method (*Berendsen et al., 1984*) and an integration step size of 1 fs. Following successful equilibration, 10 ns simulations were performed with an integration step size of 2 fs. Pressure coupling and electrostatics were turned off. Temperature was held constant at 300 K using the Berendsen coupling method. The most prevalent structures were extracted using the GROMOS clustering algorithm (*Daura et al., 2001*). Theoretical collision cross sections were calculated for each clustered structure, using the projection approximation method (as outlined in *Ruotolo et al., 2008*) and using the exact hard sphere scattering model (as implemented in the EHSSrot software; *Shvartsburg et al., 2007*).

## Circular dichroism (CD) spectroscopy

Far-UV circular dichroism (CD) spectra were recorded using a JASCO J-815 spectrometer (Jasco; Oklahoma City, OK USA) from 200 nm to 260 nm with 300 µL of 0.2 mg/mL PKM2 ion a quartz cuvette with a path length of 0.1 cm. Measurements were performed at a constant temperature of 20°C. Raw data in units of mdeg were converted to the mean residue CD extinction co-efficient, in units of M$^{-1}$ cm$^{-1}$:

$$\varepsilon_{mrw} = \frac{S \cdot MRW}{32980 \cdot c \cdot L}$$

where $c$ is the protein molar concentration, $L$ is the path length (cm), $S$ is the raw measurement of CD intensity (in units of mdeg) and $MRW$ is the molecular weight of the protein divided by the number of amino acids in the protein.

## Calculation of the allosteric coupling co-efficient for wild-type PKM2 and the AlloHub mutants

Enzyme activity measurements of the PKM2(WT) and the AlloHubMs were performed at 37 °C using a lactate dehydrogenase assay, as previously described. Initial velocities were measured over a range of phosphoenolpyruvate concentrations, with a constant concentration of 5 mM ADP. Measurements were repeated following pre-incubation of the PKM2 variant with saturating concentrations of FBP (2 µM for all variants, with the exception of R489L which was incubated with 50 mM FBP). The allosteric coupling constant ($Q$) was calculated to determine the coupling between FBP binding and catalysis, as previously described (*Reinhart, 2004*; *Reinhart, 1983*; *Reinhart, 1988*):

$$Q = \frac{K_{ia}}{K_{ia/x}}$$

where $K_{ia}$ and $K_{ia/x}$ are equilibrium dissociation constants for the binding of the substrate (*a*) in the absence or presence, respectively, of the allosteric effector *x*. When $Q > 1$, there is positive allosteric coupling between the binding of *x* to the protein and the binding of A to the substrate binding pocket. Conversely, when $Q < 1$, there is negative coupling between the A and *x* sites. Measurements were repeated after addition of 400 µM Phe to the protein variants that had been pre-incubated with FBP, and activity was measured over a range of substrate concentrations. This facilitated the calculation of the coupling constants between the Phe, FBP and substrate binding sites.

## Acknowledgements

We would like to thank all members of the Anastasiou and Fraternali laboratories for fruitful discussions, feedback and technical advice. We thank Mariana Silva dos Santos and James MacRae (Crick Metabolomics Science Technology Platform) for discussions and advice on metabolite measurements, Nicola O'Reilly (Crick Peptide Chemistry Science Technology Platform) for the synthesis of labelled peptides, and Colin Davis (Crick Proteomics Science Technology Platform) for technical assistance with proteomics. We thank Jakub Ujma and Waters Corp. for the use of the prototype SID instrument and the Wysocki Group (Ohio State University) for helpful discussions on SID tuning. AT is supported by BBSRC grants BB/L002655/1, BB/L016486/1 which is a studentship with additional financial support from Waters Corp. This work was funded by the MRC (MC_UP_1202/1) and by the Francis Crick Institute, which receives its core funding from Cancer Research UK, the UK Medical Research Council and the Wellcome Trust to DA (FC001033).

## Additional information

### Funding

| Funder | Grant reference number | Author |
| --- | --- | --- |
| Cancer Research UK | FC001033 | Dimitrios Anastasiou |
| Wellcome | FC001033 | Dimitrios Anastasiou |
| Medical Research Council | FC001033 | Dimitrios Anastasiou |
| Francis Crick Institute | FC001033 | Dimitrios Anastasiou |
| Biotechnology and Biological Sciences Research Council | BB/L002655/1 | Alina Theisen |
| Biotechnology and Biological Sciences Research Council | BB/L016486/1 | Alina Theisen |
| Medical Research Council | MC_UP_1202/1 | Dimitrios Anastasiou |

The funders had no role in study design, data collection and interpretation, or the decision to submit the work for publication.

### Author contributions

Jamie A Macpherson, Alina Theisen, Software, Formal analysis, Investigation, Methodology, Writing—original draft, Writing—review and editing; Laura Masino, Formal analysis, Investigation, Methodology, Writing—review and editing; Louise Fets, Formal analysis, Investigation, Methodology; Paul C Driscoll, Vesela Encheva, Resources, Formal analysis, Investigation, Methodology; Ambrosius P Snijders, Resources, Formal analysis, Supervision, Investigation, Methodology; Stephen R Martin, Resources, Supervision, Methodology; Jens Kleinjung, Conceptualization, Resources, Software, Methodology, Writing—review and editing; Perdita E Barran, Resources, Software, Supervision, Methodology, Writing—review and editing; Franca Fraternali, Conceptualization, Resources, Supervision, Methodology, Writing—review and editing; Dimitrios Anastasiou, Conceptualization, Supervision, Funding acquisition, Visualization, Methodology, Writing—original draft, Writing—review and editing

### Author ORCIDs

Alina Theisen http://orcid.org/0000-0002-0216-8582
Franca Fraternali http://orcid.org/0000-0002-3143-6574
Dimitrios Anastasiou https://orcid.org/0000-0002-1269-843X

### Decision letter and Author response

Decision letter https://doi.org/10.7554/eLife.45068.041
Author response https://doi.org/10.7554/eLife.45068.042

## Additional files

### Supplementary files

• Supplementary file 1. Targeted proteomics data used to quantify intracellular PKM2 concentrations.
DOI: https://doi.org/10.7554/eLife.45068.034

• Supplementary file 2. Human PKM2 as cloned in pET28a vector (His-tagged).
DOI: https://doi.org/10.7554/eLife.45068.035

• Transparent reporting form
DOI: https://doi.org/10.7554/eLife.45068.036

### Data availability

All data generated or analysed during this study are included in the manuscript and supporting files.

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
