## [Decision Letter]

Thank you for submitting your article "Functional cross-talk between allosteric effects of activating and inhibiting ligands underlies PKM2 regulation" for consideration by *eLife*. Your article has been reviewed by three peer reviewers, including Nir Ben-Tal as the Reviewing Editor and Reviewer #1, and the evaluation has been overseen by Michael Marletta as the Senior Editor. The following individual involved in review of your submission has also agreed to reveal his identity: Luke O´Neill (Reviewer #3).

The reviewers have discussed the reviews with one another and the Reviewing Editor has drafted this decision to help you prepare a revised submission.

Summary:

Experiments and computations are used to study the allosteric cross-talk between two ligand binding, FBP and Phe, on the activity of PKM2 (the glycolytic enzyme pyruvate kinase M2). FPB is an activating allosteric ligand, while Phe binding prevents maximal activation of the FBP bound PKM2.

In the first part of the manuscript the authors describe the simultaneous binding of multiple ligands to PKM2 in different cell lines and their influence on PKM2 activity. Particularly they measure the affinity of PKM2 for FBP, Phe and Ser in vitro, showing that the fractional saturation of PKM2 with FBP is close to 1 in all conditions while the fraction saturation of Phe and Ser are partial and dependent on the conditions. Thus, the activity of PKM2 is not regulated by FBP but by the amino acids. For example they show that Ser results in a decrease of the Km of PEP and an increased enzymatic activity, whereas Phe increases the Km for PEP and decreased the enzymatic activity. They next show that FBP drives PKM2 to be a tetramer, and that Phe enhances FBP-induced tetramerisation, indicating a functional synergism between the two allosteric ligands, with Phe inhibiting FBP-bound PKM2 without causing PKM2 tetramer dissociation. This second part is done with purified proteins. To understand the molecular basis of FBP-induced PKM2 allostery they conduct molecular dynamics simulations. For this they developed the AlloHubMat method to predict allosteric hub fragments from the network of dynamic correlated motions. From here they identify mutants that disrupt FBP-induced activation of PKM2 or its sensitivity to Phe. They produce and purify 7 such mutants and evaluate their activity. Indeed, they show that these mutants introduced various effects on PKM2 (as predicted by the simulation). For example, they show that residues A327 and C358 have a role in coupling the allosteric effect of Phe with that of FBP. Conversely, K305Q and F307P result in decrease in the intensity of tetramer and dimer peaks for PKM2.

Opinion:

It is an interesting problem, and the manuscript presents extensive experimental and also computational work to study the allostery between these two ligand-binding sites and the active site. Major revisions are needed to address the issues listed below.

Essential revisions:

1) The relation between the first part of the manuscript (the in cell measurements) and the other parts (MD simulations, oligomerization state and mutant analysis) is not clear. It should be further emphasized and strengthened.

2) The data on PKM2 oligomerisation are weak. More extensive cross linking analysis is needed as well as the use of gel filtration to examine the stoichiometry of PKM2, both wild type and mutant forms, to confirm what is being proposed.

3) Although the emphasis is on biochemistry, the authors need to consider other approaches, otherwise the paper will have a more limited audience. Can the authors test the physiological relevance of their findings? This could be done by reconstituting PKM2-deficient cells with some of the mutants and assessing PKM2 function in glycolysis or in its nuclear roles. More information on how the regulation reported here might be relevant to cell proliferation or cytokine production is needed. Such experiments would greatly improve the manuscript.

4) The AlloGHubMat method described is based on the GSATools that some of the authors have developed to explore allosteric communication in proteins (GSATools: analysis of allosteric communication and functional local motions using a structural alphabet", Bioinformatics. 2013;29(16):2053-5). The authors should clarify novelty, if any, compared to the 2013 publication.

5) AlloGGubMAT analysis of replica exchange molecular dynamics simulations suggests 32 allosteric positions, of which the authors identified 7 to experimentally explore. They argue that these are allosteric sites, disseminated from FBP, but it looks like the mutations are mainly around the active site, and one at FBP, which is actually a binding site. The authors should explain why they consider their results to be indicative of allostery.

6) The authors should clearly indicate where the Phe binding site is. (We assume that it binds in the same site as Ser in another PDB structure.)

7) They should also include figures, perhaps also PyMOL sessions or something, to highlight the known binding/active sites and the location of the putative allosteric sites to make it clear that the orthosteric and allosteric sites are separated from each other.

8) The difference between the apo and FBP bound mutual information gives positions within the subunit/monomer. What about inter-subunit cooperativity or intersubunit allosteric ligand cooperativity? The absence of large conformational changes doesn't exclude the inter-subunit dynamics.

9) The manuscript states that apo and holo states are similar and there are no large tertiary or quaternary structure changes, with no evidence of global conformational changes. Is it so? Perhaps there are no large global conformational changes, but it seems that there is a global rearrangement. A simple alignment of the two pdb structures (3bjt and 3bjf) shows some global structural adjustment from the apo to PFB bound states that possibly may come up with some dynamic changes that could involve an allostery among monomers within the tetramer.

10) The reference to the available structures is confusing. 3u2z in the method and 3bjf in the simulation table are referred for the FBP bound structures. Since the crosstalk between Phe and FBP is examined, why shouldn't the Phe bound structure be used? Or at least dock the Phe? (Maybe it was done and we missed it?) And PDB 4b2d, where Ser and FBP are bound, should be referred to, perhaps used. Are there other relevant structures?

11) In the Introduction the authors refer to low affinity T- and high affinity R-state tetramers. However, they don't relate these to the known structures. Maybe, the transition between these two states would also be informative?

12) In the beginning, the manuscript says that it is possible that the allostery has enthalpic motion: "we found no significant difference in the time-dependent configurational entropy of the PKM2 apo and PKM2FBP simulated trajectories". However, then it talks about positional entropy? And never get back to enthalpy in discussions. All of these discussions are not that clear. What is the conclusion in terms of enthalpy vs. entropy?

---

## [Author Response]

Essential revisions:1) The relation between the first part of the manuscript (the in cell measurements) and the other parts (MD simulations, oligomerization state and mutant analysis) is not clear. It should be further emphasized and strengthened.

We agree with the reviewers that we should have been clearer in explaining the connection between intracellular metabolite concentration measurements and our biophysical studies, which we hope we have achieved in the revised manuscript. In brief, in the revised Introduction (last paragraph), we clarify that, in vitro several ligands have been shown to regulate PKM2 both alone and in combination; we, therefore, selected PKM2 as a suitable model system to study the problem of allosteric signal integration. However, we first needed to determine whether simultaneous ligand binding is likely to reflect a scenario that can also occur in cells. For example, if all ligands were found at sub-saturating amounts, it would have been conceivable that they could bind to PKM2 independently of each other. Our metabolomics measurements indicate that Phe is unlikely to bind to and regulate apo-PKM2. This is because FBP is at such high concentrations relative to its binding affinity, that is predicted to saturate PKM2. Given that FBP alters the mode of action of Phe (Figures 3B, 4A), our intracellular metabolite measurements provide further strong support for the suitability of PKM2 to address this fundamental question in allosteric control.

2) The data on PKM2 oligomerisation are weak. More extensive cross linking analysis is needed as well as the use of gel filtration to examine the stoichiometry of PKM2, both wild type and mutant forms, to confirm what is being proposed.

Cross-linking and gel filtration have each been historically very useful in studying protein oligomerisation, however, native mass spectrometry (MS) is a very powerful technique that offers additional advantages and information that are critical to the aims of our study. A key point is that gel filtration separates proteins according to their hydrodynamic radius, which is influenced by both the protein’s mass and shape, the relative contribution of which cannot be readily distinguished. The ability of MS to directly determine, at high resolution, protein mass (Figure 4A) and shape (in ion mobility mode, Figure 4D), provided us with unprecedented insights as detailed below.

a) Native MS offers sufficiently high mass resolution to directly observe allosteric ligand binding on PKM2 and stoichiometry, which is impossible by cross-linking and gel filtration. This capability allowed us to distinguish PKM2^apo^ tetramers from tetramers with 1, 2, 3 and 4 bound FBP molecules (Figure 4C), thereby providing the following critical insights:

- Given the presence of residual FBP (Figure 2—figure supplement 1, which, as discussed in our manuscript, is likely to impact the conclusions of many other studies) it has remained unclear whether PKM2 tetramers in PKM2^apo^ preparations can form in the complete absence of FBP. Native MS allowed us to unequivocally demonstrate that PKM2 tetramers can form in complete absence of FBP (Figure 4C).

- Furthermore, with surface-induced dissociation (SID), we could determine that FBP-bound PKM2 tetramers are more stable than PKM2^apo^. This is also not possible by gel filtration and cross linking.

- Importantly, using native MS we could show that both FBP and Phe can simultaneously bind to PKM2 (Figure 4—figure supplement 4). We were also able to obtain dynamic measurements of tetramer formation in the presence of FBP alone or FBP+Phe, which demonstrated that co-occupancy of FBP and Phe promotes tetramerisation (Figure 4E), further supporting the idea that tetramerisation and activity can be uncoupled.

b) Ion mobility MS enabled us to identify for the first time, to the best of our knowledge, the exact configuration of the PKM2 dimer (Figure 4—figure supplement 1). Such information proved essential for interpreting the behaviour of two of the mutants (K305 and K307 at the interface of the identified PKM2 dimer) and would have been impossible to obtain directly by cross-linking and/or gel filtration.

c) Our proteomic measurements indicate that intracellular PKM2 concentration ranges between 2-2.5 μM (Figure 2) and in our native MS experiments PKM2 is used at 5-20 μM. Therefore, the concentration of PKM2 used to study the effect of allosteric ligand binding on oligomerisation is close to that found in cells. Although, clearly, there are other cellular factors, such as molecular crowding, that may influence oligomerisation and that cannot be faithfully mimicked in vitro, the concentration of the protein is a major determinant for its oligomerisation and therefore important to be as close as possible to that found in cells. In contrast, even starting with a concentrated protein preparation, in the gel filtration column the protein undergoes several hundred-fold dilution that cannot be well-controlled. This can be overcome by a combination of cross-linking and gel filtration, however, native MS achieves this in a single experimental setup.

In conclusion, the above considerations highlight that native MS provides all the information required to address the questions in our study that could be obtained with cross-linking, gel filtration or combination thereof. In addition, native MS has several additional advantages that overcome the key limitations of these, otherwise useful, techniques.

3) Although the emphasis is on biochemistry, the authors need to consider other approaches, otherwise the paper will have a more limited audience. Can the authors test the physiological relevance of their findings? This could be done by reconstituting PKM2-deficient cells with some of the mutants and assessing PKM2 function in glycolysis or in its nuclear roles. More information on how the regulation reported here might be relevant to cell proliferation or cytokine production is needed. Such experiments would greatly improve the manuscript.

We agree with the reviewers that such biological insights would be highly interesting and, indeed, we have initiated studies to this end. However, to obtain meaningful conclusions from such investigations is a major project on its own right that would require significant time to carry out for reasons that are outlined below.

While for our present study we could *acutely* (1h treatment, see Materials and methods) modulate intracellular amino acid concentrations by changing media composition (as in Figure 2C), assessing effects of varying amino acid concentrations on *long-term* cellular function, such as proliferation, over a period of days is more challenging. This is because cells may adapt to extracellular amino acid availability (e.g. by activating autophagy) and have varying amino acid synthesis and uptake capacity [well documented for serine, e.g. see Locasale et al. (2011) Nature Genetics, 43(9):869–874]. Therefore, for the reviewers’ proposed experiments, we would need to experimentally validate suitable cell lines that are amenable to long-term culture in varying amino acid concentrations and/or genetically manipulate simultaneously at least two amino acid synthesis pathways.

Finally, PKM2 activity may influence, as the reviewers suggest, non-metabolic functions; so, linking any effects of varying amino acid concentrations to cellular phenotypes would necessitate significant additional work. For example, assessing cytokine production, as suggested, would require us to establish new cellular systems (e.g. macrophage culture and stimulation) that are not readily available in our lab.

For these reasons, we retained the focus of the revised manuscript on the important problem of allosteric signal transmission and integration. Allostery is a ubiquitous process and how allosteric signals from multiple ligands are integrated is a fundamental question likely to impact many allosterically regulated proteins. We therefore feel that our study is highly relevant as it stands. In the revised manuscript, we further emphasise that we use PKM2 as a model to study this fundamental question in allosteric control and have clarified the reasons why PKM2 is suitable for studying this question. In the concluding paragraph of our manuscript (subsection “Multiple allosteric inputs in the context of intracellular concentrations of allosteric effectors and other modifications”), we also clarify that our study provides a road map for generating the necessary tools towards future studies along the lines of the reviewers’ suggestions.

4) The AlloGHubMat method described is based on the GSATools that some of the authors have developed to explore allosteric communication in proteins (GSATools: analysis of allosteric communication and functional local motions using a structural alphabet", Bioinformatics. 2013;29(16):2053-5). The authors should clarify novelty, if any, compared to the 2013 publication.

We agree and in the revised manuscript we explicitly state how AlloHubMat compares to GSATools (subsection “AlloHubMat reveals residues that mediate a cross-talk between FBP- and Phe-induced allosteric regulation”, second paragraph). In detail: the GSATools software employed the same method as AlloHubMat for detecting Mutual Information between (distant) sites. However, GSATools was limited to the comparison of two trajectories per analysis. AlloHubMat overcomes this limitation through diagonalisation of Mutual Information matrices to eigensystems. Comparing the overlap of the spaces spanned by the eigenvectors allowed us to cluster together different trajectories that showed high eigenspace overlap in the leading eigenvectors, because they represent similar substates in terms of their Mutual Information. Moreover, those Mutual Information matrices can be averaged and substates can be compared by subtracting their averaged Mutual Information matrices. Sites showing large differential signals are potential allosteric hub residues.

AlloHubMat is therefore a far more versatile software, because multiple trajectories and the substates they contain can be analysed in one framework. Additionally, we have re-written the user interface in R with calls to fast C routines for demanding computations (Kabsch superpositioning for encoding Mutual Information between all pairs of alignment columns) and we are currently revising the code of the AlloHubMat method for public release.

5) AlloGGubMAT analysis of replica exchange molecular dynamics simulations suggests 32 allosteric positions, of which the authors identified 7 to experimentally explore. They argue that these are allosteric sites, disseminated from FBP, but it looks like the mutations are mainly around the active site, and one at FBP, which is actually a binding site. The authors should explain why they consider their results to be indicative of allostery.

We would first like to clarify that the coverage of our validated allosteric sites is greater than that implied above. Our analysis identified only 10 allosteric hub fragments (AlloHubFs). Each AlloHubF is encoded by a letter of the structural alphabet and therefore comprises 4 amino acids (Figure 6—figure supplement 2). AlloHubMat does not indicate which of the amino acids within the AlloHubFs carry the allosteric information. Therefore, to decide which residue(s) within each AlloHubF to mutate to disrupt allostery, we reasoned that, as FBP-induced allostery is well-conserved, residues within AlloHubFs that are critical for FBP-induced allostery are more likely to be conserved. To this end, we aligned 5381 PK orthologues and identified the most highly conserved residue within each AlloHubF, which we then mutated to one or more residues with a chemically different side chain. When possible, we selected substitutions that are found in PK orthologues (e.g. Ala is frequently found in position 358, although in the human protein it is a Cys), reasoning that such residues are likely well-tolerated and less likely to perturb the overall structure of PKM2. In total, we generated 23 (not 32, a typographical error now corrected in the revised manuscript) mutants (AlloHubMs), listed in Author response image 1, providing some redundancy of mutants compared to the number of AlloHubFs. Among these mutants, only the 7 highlighted in green in Author response image 1, expressed well and yielded protein preparations of good quality, which we used for further experiments. Nevertheless, these 7 mutants provided a 78% coverage of the 10 AlloHubF (Hubs 5 and 6 overlapping).

**Author response image 1. respfig1:** Summary of AlloHub mutants generated for this study.

Regarding whether the identified residues are allosteric, all AlloHubF residues mutated were selected based on a stringent statistical selection (Figure 5C) of their increased correlated motions upon FBP binding in MD simulations of PKM2 without substrates or amino acid ligands. With the exception of R489, all AlloHub sites are distal from the FBP binding pocket, therefore, by definition, they are allosterically coupled to FBP.

Among the 7 mutants, F244 and A327 are close to the catalytic pocket, R489 is in the FBP binding pocket, and C358 is close to the amino acid binding pocket but does not contact Phe (Author response image 1). We agree that, given the proximity of some of the AlloHubF sites to the catalytic pocket and ligand binding pockets, some mutants might influence catalysis independently of any allosteric effects. This situation may result from the mutations affecting (i) catalysis, (ii) FBP binding, or (iii) amino acid binding.

i) To assess the contribution of AlloHub residues to FBP-induced allostery independently of any direct effects on catalysis or ligand binding, we used the formalism of *allosteric coupling coefficient Q* (Figure 6B, C) [detailed in Reinhart, 2004, and more recently discussed in Carlson and Fenton, 2016 – see p 1913 therein]. Q is the ratio of the substrate (PEP) binding constants with and without an allosteric ligand, and therefore it allows us to assess the allosteric effect of these ligands independently of the absolute values of either binding constants. This approach allowed us to assess the contribution of a residue to allostery even if a mutation influences substrate binding.

ii) With the exception of R489L, a site at the FBP pocket that, as expected, has lower affinity for FBP, none of the other mutants significantly affect FBP binding (Figure 6—figure supplement 4 and Table 7), excluding disrupted FBP binding as the reason for altered FBP-induced allostery.

iii) On the other hand, some of the AlloHub mutants may affect amino acid binding. This caveat would be most pertinent for mutants K305Q, F307P, A327S and C358A that show impaired inhibition by FBP (Figure 6B, C). However, impaired Phe binding is unlikely because all four mutants are distal from the Phe binding pocket and the closest one, C358, does not make any contacts with Phe (Author response image 2).

These considerations strongly support our conclusion for an allosteric role of the identified sites.

**Author response image 2. respfig2:** AlloHub mutant locations relative to catalytic and ligand binding pockets. (**a**) PDBID:3u2z (PKM2+FBP), (**b**) PDBID:6gg4, (**c**) Phe binding pocket in 6gg4 plotted using LigPlot [Wallace et al. (1996). Protein Eng., 8:127-134].

6) The authors should clearly indicate where the Phe binding site is. (We assume that it binds in the same site as Ser in another PDB structure.)

We agree with the reviewers and in the revised manuscript we have included this information as a new Figure 1, cited in the Introduction. In new Figure 1, we indicate more clearly that, as the reviewers rightly point out, Phe binds in the same pocket as Ser. We also mention this explicitly in the text (subsection “Phe inhibits FBP–bound PKM2 without causing PKM2 tetramer dissociation”, second paragraph).

7) They should also include figures, perhaps also PyMOL sessions or something, to highlight the known binding/active sites and the location of the putative allosteric sites to make it clear that the orthosteric and allosteric sites are separated from each other.

As for point 6, above, this information is now included as a new Figure 1, cited in the Introduction (fourth paragraph) and in the corresponding text.

8) The difference between the apo and FBP bound mutual information gives positions within the subunit/monomer. What about inter-subunit cooperativity or intersubunit allosteric ligand cooperativity? The absence of large conformational changes doesn't exclude the inter-subunit dynamics.

We have computed the difference Mutual Information (MI) matrix between the apo- and FBP-bound states of PKM2 for subunits A and B in a tetramer simulation. This extends the analysis presented in the manuscript, which was performed on single subunits only of a tetramer simulation. Being approximately normally distributed, the MI difference values were converted into probabilities and adjusted using the Benjamini-Hochberg method for multiple testing. Peak padj values were extracted from intra- and inter-subunit matrices at the positions of the hub residues selected in the manuscript. Author response table 1, below, summarises the padj values in negative log-transformed form. A lower peak cut-off of 18 was chosen to match the criteria for identifying hub residues in the manuscript.

**Author response table 1. resptable1:** Comparison of intra- and inter-subunit Mutual Information signal strength for allosteric hubs identified in this study.

Hub	Intra	Inter
H1	42.9	-
H2	35.4	-
H3	22.3	-
H4	88.8	18.2
H5_6	18.8	21.5
H7	20.5	19.7
H8	21.4	18.7
H9	22.3	-
H10	29.9	-

The inter-domain signalling is weaker than the intra-subunit signalling for most hubs, as expected. However, the hubs near the large interaction surface A-A’ between subunits A and B, namely H5_6 (see Figure 5E), show a strong inter-subunit coupling, while the more distant hubs H4, H7 and H8 show a slightly weaker coupling. This brief analysis shows that inter-subunit signalling is detectable by our method, but only for some of the hub residues tested experimentally. A complete analysis of inter-subunit signalling and comparison with intra-subunit signalling would require a separate study, not least for the design of experiments to validate the predicted molecular effects of intra- and inter-subunit signalling.

9) The manuscript states that apo and holo states are similar and there are no large tertiary or quaternary structure changes, with no evidence of global conformational changes. Is it so? Perhaps there are no large global conformational changes, but it seems that there is a global rearrangement. A simple alignment of the two pdb structures (3bjt and 3bjf) shows some global structural adjustment from the apo to PFB bound states that possibly may come up with some dynamic changes that could involve an allostery among monomers within the tetramer.

We agree with the reviewers that the phrasing we used in the original manuscript was imprecise and warranted further clarification. Our ion mobility MS measurements (Figure 4D) show reproducible differences between the collisional cross sections of PKM2^apo^ and PKM2^FBP^, indicating conformational changes upon FBP binding (which may relate to the structural adjustment upon FBP binding the reviewers refer to). Nevertheless, the magnitude of these differences is small. This is further supported by the analysis of the MD trajectories that show minimal changes in solvent accessibility and protein volume upon FBP binding (Figure 5—figure supplement 1). Therefore, with 'no global changes' we meant that no *drastic* structural changes occur. Regardless their magnitude, such rearrangements are associated with a higher Mutual Information between distal sites across the structure, which we go on to identify with the AlloHubMat algorithm. In the revised manuscript (subsection “Molecular dynamics simulations reveal candidate residues that mediate FBP–induced PKM2 allostery”, first paragraph), we have incorporated the reviewers’ wording suggestion and clarify this point as outlined here.

10) The reference to the available structures is confusing. 3u2z in the method and 3bjf in the simulation table are referred for the FBP bound structures. Since the crosstalk between Phe and FBP is examined, why shouldn't the Phe bound structure be used? Or at least dock the Phe? (Maybe it was done and we missed it?) And PDB 4b2d, where Ser and FBP are bound, should be referred to, perhaps used. Are there other relevant structures?

We thank the reviewers for pointing out the discrepancy between the reported pdb identifiers. Indeed, the structure 3bjf has been referenced in Table 6 in error. For our simulations of FBP-bound PKM2 we used 3u2z as stated, correctly, in the Materials and methods section. This error in Table 6 has been corrected in the revised manuscript. Furthermore, in new Figure 1, we present other relevant structures incl. that of Ser bound to PKM2.

We have not used amino acid-bound PKM2 structures for our theoretical analyses for reasons outlined below. Our experimental data indicated that amino acids alter the mode of action of FBP (Figures 3 and 4), providing empirical evidence that amino acid binding may interfere with the allosteric pathway used by FBP. For this reason, we first identified the allosteric pathway of FBP, then investigated if residues along this pathway can account for the observed cross-talk with amino acid binding. Our analyses revealed the existence of residues (A327 and C358) along the FBP allosteric pathway that can be mutated without affecting the ability of FBP to activate PKM2, but prevent the allosteric effect of Phe on PKM2 activity. The FBP allosteric pathway, therefore, reaches the amino acid binding pocket, and the mutants designed on the basis of the theoretical work show that the allosteric mechanism was sampled even without bound amino acids. In that respect, our current results are self-consistent. Finally, we agree that studies with amino acids bound would be desirable. However, to simulate and analyse a system with four subunits, ± FBP, ± Ser or ± Phe would warrant a separate study because of the combinatorial possibilities to account for and the sheer amount of computational work.

11) In the Introduction the authors refer to low affinity T- and high affinity R-state tetramers. However, they don't relate these to the known structures. Maybe, the transition between these two states would also be informative?

We introduce the existence of the R- and T-state tetramers here only as a reference to alternate high- and low-affinity states that tetrameric PKM2 adopts for its substrate, when bound to activating or inhibitory ligands, respectively. We have previously shown that, in simulations of PKM2 monomers, FBP induces a substantial movement of the B-domain that results in a closed conformation, compared to apo-PKM2 monomers. This B-domain closure is associated with the catalytic His78 adopting a side-chain conformation similar to that seen in PKM2 co-crystallised with activating ligands, i.e. the R-state (Gehrig et al., 2017). However, in the context of PKM2 tetramer simulations, we observe only modest gross structural re-arrangements upon ligand binding. This is further supported by empirical evidence from our ion mobility MS experiments in this manuscript. In the absence of large global configurational rearrangements, we show evidence for a network of correlated backbone motions, which provides the basis for coupling between the FBP and substrate binding pockets. To the extent that FBP-induced changes in coupling between distal sites reflect a T-like or R-like state, our MD simulation analyses essentially show this transition and address the reviewers’ suggestion.

12) In the beginning, the manuscript says that it is possible that the allostery has enthalpic motion: "we found no significant difference in the time-dependent configurational entropy of the PKM2 apo and PKM2FBP simulated trajectories". However, then it talks about positional entropy? And never get back to enthalpy in discussions. All of these discussions are not that clear. What is the conclusion in terms of enthalpy vs. entropy?

We agree with the reviewers that the statements about the entropy we calculate were unclear. This is because we were referring to entropy at two different types of resolution and therefore different in nature. The following points should clarify that:

a) We compute information-theoretic properties of the molecular trajectory after encoding it into a state sequence (i.e. states of the structural alphabet). Among those properties is the Shannon entropy per position, which is the entropy of all states observed at one structural position over the considered time period. The Shannon entropy is the information theoretic metric of the state model that corresponds to the root-mean-square-fluctuation of the backbone. We have shown in our paper introducing the structural alphabet (Pandini et al., 2010) that this correlation was consistently high. Therefore, *positional* entropy refers to this notion of backbone fluctuation.

b) The *total* entropy of the simulated protein was computed using the Schlitter formula. That is an entropy term with a physico-chemical foundation, which takes into account the vibrational modes of the protein. However, the statement about the role of the total entropy in the observed allosteric process was imprecise and it has been removed together with the accompanying figure (panel C in Figure 5—figure supplement 1) (subsection “Molecular dynamics simulations reveal candidate residues that mediate FBP–induced PKM2 allostery”, first paragraph). The enthalpic and entropic differences accompanying allosteric processes are usually small compared to the total enthalpy and entropy of the system. Therefore, we focus our analysis on relative differences; the total quantities are less suitable descriptors of the allosteric process. We thank the reviewers for pointing out this incoherence.

Our computational method is designed to capture correlations and measure them as information theoretic signals. Other allostery methods compute explicitly enthalpic, entropic or free energy/enthalpy contributions to the allosteric process. It would be very interesting to integrate both approaches, but at the moment we have not yet implemented the thermodynamic analysis in our framework.